# HIV-1 uncoating location dictates sites of integration

Ryan C. Burdick [1] ✉, Sean C. Patro[2], Ellie Bare [1], Rokeya Siddiqui[1], Krista A. Delviks-Frankenberry [1], Olga A. Nikolaitchik[3], Stephen H. Hughes [4], Xiaolin Wu [2], Wei-Shau Hu[3] & Vinay K. Pathak [1] ✉

HIV-1 cores enter the nucleus and undergo capsid disassembly (uncoating) near their integration site. Although most viral cores are localized to nuclear speckles (NSs), the spatial relationship between the uncoating site and integration site remains unclear. Here, using fluorescently labeled HIV-1 cores and NS markers, we show that uncoating predominantly occurs within NSs. Treatment of infected cells with capsid inhibitors PF-3450074 (PF74) or lenacapavir (LEN) after nuclear entry induced rapid disruption of interactions between capsid and cleavage and polyadenylation specificity factor 6 (CPSF6) followed by exit of HIV-1 cores from NSs, indicating that CPSF6 binding is required to retain the viral cores in the NSs. Treatment with PF74 or LEN led to core disruption and appearance of transcriptionally active proviruses further from the NSs compared to viral cores that uncoated in the NSs in untreated cells. This spatial shift correlated with reduction in integration into gene-rich, transcriptionally active speckle-associated chromatin domains, the preferred sites of HIV-1 integration, and increased integration into gene-sparse lamina-associated domains located away from the nuclear envelope. These findings demonstrate that the HIV-1 uncoating site is a key determinant of integration targeting, and that capsid inhibitors can misdirect integration by relocalizing uncoating to outside of NSs.

Human immunodeficiency virus type 1 (HIV-1), a retrovirus that infects CD4[+] immune cells, is the causative agent of the acquired immunodeficiency syndrome (AIDS) pandemic. During viral maturation, HIV-1 protease cleaves the Gag and GagPol polyprotein precursors into fully processed viral proteins, leading to the formation of a mature HIV-1 capsid—a closed, conical shell composed of ~250 hexamers and exactly 12 pentamers of the capsid protein (CA)[1]. The HIV-1 capsid, which encloses the viral RNA genome along with the enzymes reverse transcriptase (RT), integrase (IN) and other proteins to form the viral core, serves as a protective container that facilitates efficient reverse

transcription[2] and shields the viral DNA from innate immune sensors[3–5]. For integration to occur, the capsid must disassemble—a process known as uncoating—allowing integration of the newly synthesized double-stranded viral DNA into the host genome to form a provirus. Elegant studies performed over two decades ago showed that the CA is functionally required for infection of non-dividing cells[6] and that optimal capsid stability is required for efficient reverse transcription[7]; however, the timing and location of HIV-1 uncoating remained elusive until recently. Our live-cell imaging studies showed that the viral cores that lead to productive infection remain intact as

[1]Viral Mutation Section, HIV Dynamics and Replication Program, Center for Cancer Research, National Cancer Institute, National Institutes of Health, Frederick, MD, USA. [2]Cancer Research Technology Program, Frederick National Laboratory for Cancer Research, Frederick, MD, USA. [3]Viral Recombination Section, HIV Dynamics and Replication Program, Center for Cancer Research, National Cancer Institute, National Institutes of Health, Frederick, MD, USA. [4]HIV Dynamics and Replication Program, Center for Cancer Research, National Cancer Institute, National Institutes of Health, Frederick, MD, USA. ✉e-mail: burdickrc@mail.nih.gov; pathakv@mail.nih.gov

they enter the nucleus and uncoat only after reverse transcription is complete, in proximity to their eventual integration sites[8,9]. Consistent with our observations, it was demonstrated that reverse transcription is completed after nuclear entry[10–12] and multiple recent studies have reported the presence of intact or largely intact HIV-1 capsids within the nucleus[13–17].

The HIV-1 capsid contains -1500 phenylalanine-glycine (FG)-binding pockets, each located at the interface between CA monomers within a hexamer[18]. These pockets interact with FG dipeptide motifs found in host nucleoporins, facilitating early steps of infection, such as docking at the nuclear pore complex (NPC) and nuclear import[19–23]. Following nuclear entry, most HIV-1 cores traffic to nuclear speckles (NSs)[12,24–26]. NSs are dynamic, membraneless nuclear compartments located in the interchromatin space and closely associated with transcriptionally active chromatin[27]. At the molecular level, NSs are phase-separated nuclear bodies organized by two large, intrinsically disordered region-rich proteins, SON and SRRM2[28], and are enriched in pre-mRNA splicing factors and other RNA-processing proteins[29]. A key host factor involved in both the nuclear import of HIV-1 cores and their subsequent targeting to NSs is cleavage and polyadenylation specificity factor 6 (CPSF6)[8,30,31], a component of the cleavage complex involved in mRNA 3′ end processing. CPSF6 binds directly to capsids at NPCs and within the nucleus via a central prion-like low-complexity region that contains a single FG motif[32,33]. While the precise mechanism by which CPSF6 mediates nuclear import and NS localization remains unclear, the phase separation activity associated with its C-terminal mixed-charge domain has been implicated[34]. However, this domain can be functionally replaced by certain heterologous nuclear localization signals that, despite lacking mixed-charge domains, are sufficient to restore both nuclear import and NS targeting[35].

Provirus formation enables robust viral gene expression and establishment of a persistent viral reservoir[36]. Viral DNA integration is directed by interactions between viral integrase and host protein lens epithelium-derived growth factor (LEDGF)/p75[37], which preferentially targets viral DNA to transcriptionally active, highly spliced genes located within gene-dense chromatin regions[38,39]. Integration frequently occurs within speckle-associated domains (SPADs), regions of active chromatin located adjacent to NSs[25,40]. Mutations in capsid that disrupt its interaction with CPSF6 impair nuclear import, resulting in uncoating at the NPC and misdirected integration into lamina-associated domains (LADs)[41,42], gene-poor heterochromatin-rich regions that constitute transcription-repressive chromatin often associated with the nuclear envelope[43]. These findings underscore the importance of proper viral core trafficking and localization to NSs for integration into SPADs, the preferred targets of HIV-1 integration. However, the spatial relationship between the HIV-1 uncoating site, NSs, and the integration site remains poorly understood.

Pharmacological targeting of the HIV-1 capsid has emerged as a potent antiviral strategy[44]. PF74 and LEN are peptidomimetic capsid inhibitors that bind to the FG-binding pocket and inhibit HIV-1 replication with micromolar and subnanomolar half-maximal effective concentration ($EC_{50}$) values, respectively[45–48]. Both PF74 and LEN alter capsid stability in ways that block multiple early replication steps including reverse transcription and nuclear import[48–50]. Both inhibitors can disrupt viral core integrity; however, LEN stabilizes the capsid lattice whereas PF74 promotes capsid disassembly[8,51–53]. Despite these opposing effects on capsid lattice stability, treating cells with either inhibitor after nuclear import and completion of reverse transcription can trigger uncoating and release of viral DNA (vDNA) capable of integrating into the host genome[54]. At high concentrations, PF74 and LEN can displace host factors that also bind to the FG-binding pocket, including CPSF6[16,33]. However, the impact of capsid inhibitors on viral core localization to NSs and integration targeting have not been elucidated.

Here, we examined the spatiotemporal relationship between the location of HIV-1 uncoating, NSs, and the site of integration. Live-cell imaging of target cells expressing a fluorescent NS marker infected with dual-labeled HIV-1 indicated that loss of viral core integrity and uncoating primarily occurs within NSs. The addition of capsid inhibitors after viral cores localized to NSs induced rapid dissociation of CPSF6 from the viral cores, exit of the viral cores from NSs, and uncoating outside NSs. Proviruses resulting from the capsid inhibitor-induced uncoating were located further from NSs than proviruses in untreated control cells, which correlated with decreased integration into SPADs and increased integration into LADs located away from the nuclear envelope. Overall, these findings highlight the critical role of the location of uncoating in determining HIV-1 integration site selection and demonstrate how capsid inhibitors can misdirect integration by altering the site of uncoating to outside of NSs.

## Results

To investigate the spatial relationship between HIV-1 uncoating and NSs, we created a HeLa-derived cell line stably expressing HALO-tagged serine/arginine-rich splicing factor 1 (HALO-SRSF1), a protein enriched in NSs[55] (Fig. 1a). HALO is a genetically encoded protein tag that enables covalent labeling of proteins of interest with synthetic fluorescent ligands, facilitating their visualization in cells. These cells were infected with HIV-1 labeled with GFP-CA, which incorporates into the capsid lattice during virus maturation[8,52], and content marker HALO tagged with the far-red dye JF646 (cmHALO), a fluid-phase protein that is passively packaged within the viral core during maturation (Supplementary Fig. 1a–d). This dual-labeling strategy allows visualization of individual HIV-1 cores during uncoating (Fig. 1b). The loss of the JF646-tagged cmHALO indicates loss of viral core integrity via the formation of holes in, or the rupture of the capsid lattice, similar to other fluorescent protein fluid phase markers[9,13,56,57], while loss of GFP-CA reflects disassembly of the capsid lattice[52], as previously described[13,16]. Using live-cell imaging, we tracked 170 nuclear viral cores between 10 and 12 hours post-infection (Fig. 1c), which is when uncoating typically occurs inside the nucleus[8,9,54]. Most GFP-CA-labeled viral cores that were inside the nucleus at 10 hours post-infection were cmHALO+ and localized to bright HALO-SRSF1 signals, indicating that most nuclear viral cores remained intact and localized to NSs (-85%; Fig. 1d). We identified 33 cores that ruptured and lost cmHALO during the observation period. Of these, 31 also subsequently lost GFP-CA, indicating that they had uncoated. The remaining two cores lost cmHALO near the end of the movie but did not lose GFP-CA and were therefore excluded from further analysis. Quantification of the fluorescence intensities revealed that -50% loss of the GFP-CA signal occurred approximately -8 min after the loss of the cmHALO signal (Fig. 1e), indicating that capsid lattice disassembly occurs rapidly after loss of viral core integrity. In contrast, 30 randomly selected viral cores that did not uncoat exhibited minimal loss of cmHALO and GFP-CA, indicating negligible photobleaching during the imaging period (Supplementary Fig. 1e). Next, we determined the location of uncoating relative to NSs and observed that all 31 viral cores lost cmHALO inside NSs, and that most of these viral cores (84%) also lost all the GFP-CA signal inside NSs (Fig. 1f; example shown in Fig. 1g and Supplementary Movie 1). A minority (16%) of viral reverse transcription/preintegration complexes (RTCs/PICs) exited the NSs following loss of cmHALO (core rupture) and substantial loss of GFP-CA (uncoating) and subsequently lost all detectable GFP-CA signal outside the NSs (Fig. 1f; example shown in Fig. 1h and Supplementary Movie 2). Collectively, these findings suggest that HIV-1 core integrity loss, and subsequent capsid disassembly predominantly occurs within NSs.

Live-cell imaging was used to assess the spatiotemporal relationship between PF74- or LEN-induced uncoating and NSs (Fig. 2a, b, respectively). Infected cells were treated with 10 µM PF74 or 100 nM LEN after viral cores had entered the nucleus and localized to NSs. We

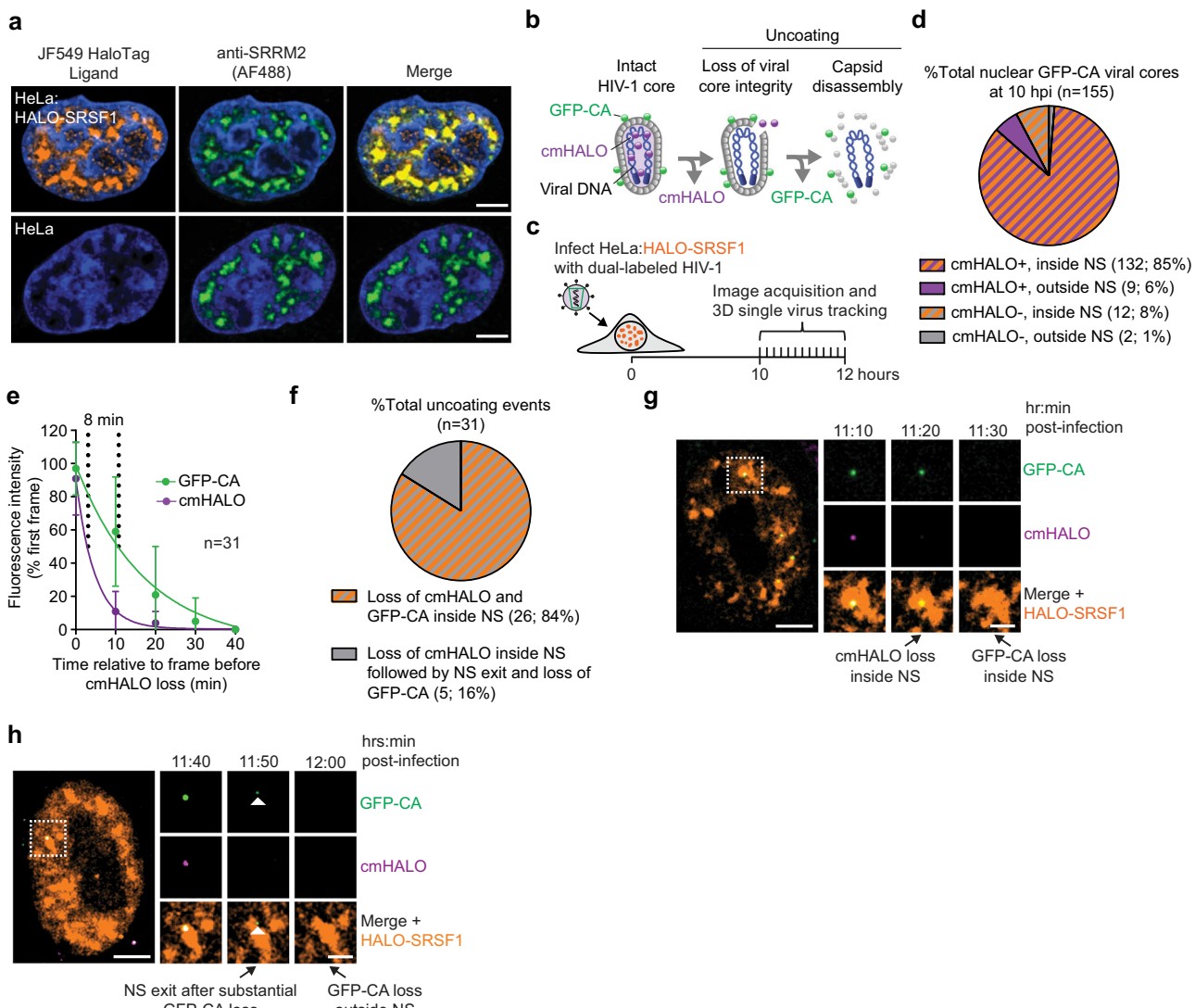

**Fig. 1 | HIV-1 uncoating occurs within nuclear speckles. a** HALO-tagged SRSF1 localizes to nuclear speckles (NSs). HeLa cell line expressing HALO-tagged SRSF1 (top row; representative example from 55 cells) or plain HeLa cells (bottom row; representative example from 62 cells) were stained with JF549. Nuclear speckles were identified by immunostaining for the NS marker SRRM2. Cells were stained with DAPI to visualize nuclei. Scale bars, 5 μm. **b** Schematic of an HIV-1 core labeled with GFP-tagged capsid protein (GFP-CA; green spheres) and a content marker HALO conjugated to JF646 (cmHALO); purple shade and spheres). Loss of cmHALO indicates disruption of core integrity, while loss of GFP-CA reflects disassembly of the capsid lattice. **c** Schematic of live-cell imaging assay to visualize NSs and uncoating of nuclear HIV-1 cores. HeLa:HALO-SRSF1 cells were infected with dual-labeled HIV-1 particles; images were acquired 10-12 hours post infection to enable 3-dimensional (3D) single virus tracking. **d** Percentage of nuclear GFP-CA–labeled viral cores that are cmHALO⁺ or cmHALO⁻ and their localization to NSs based on colocalization with HALO-SRSF1 at 10 h post-infection (155 viral cores analyzed from 3 independent experiments). **e** A total of 170 viral cores were tracked in live-cell movies, of which 31 viral cores uncoated; quantification of the sequential loss of cmHALO and GFP-CA is shown. Fluorescence intensities of the GFP-CA and cmHALO signals were normalized to the first frame and are plotted relative to the frame immediately before initial cmHALO signal loss. Curve fitting was applied to determine the time at which ~50% signal loss occurred for each fluorophore. Error bars represent standard deviation. **f** Location of cmHALO and GFP-CA loss relative to NSs. **g** Representative viral core (1 of 26) that lost cmHALO and GFP-CA within the NS. **h** Representative viral core (1 of 5) that lost cmHALO within the NS, exited the NS following substantial loss of GFP-CA, and then completely lost GFP-CA. White arrow indicates low GFP-CA signal at the time of NS exit. In (**g**) and (**h**), scale bars, 5 μm; insets, 2 μm. In **d** and **f**, numbers and percentages are shown in parentheses. Source data are provided as a Source Data file.

selected 100 nM LEN (~2500× EC₅₀) because the kinetics of viral core integrity loss at this concentration was comparable to those observed with 10 μM PF74 (~100× EC₅₀)[52]. Note that an extended imaging window was necessary to capture the slower dynamics of NS exit following LEN treatment compared to PF74 treatment (results described below). In contrast to DMSO-treated controls, most viral cores ruptured and lost the cmHALO signal following PF74 treatment (39 of 48; 81%; Supplementary Fig. 2a). Of these, 46% lost cmHALO inside NSs, while 54% lost cmHALO after exiting NSs (Fig. 2c). Notably, in contrast to substantial GFP-CA loss inside NSs during uncoating in untreated cells, no GFP-CA loss was observed inside NSs (Fig. 2d). All viral cores that

ruptured during the observation period exited NSs without loss of GFP-CA (Fig. 2e), indicating that uncoating likely occurred outside the NSs. Representative examples of viral cores that lost cmHALO before or after exiting a NS are shown in Fig. 2f and Supplementary Movie 3, and Fig. 2g and Supplementary Movie 4, respectively. After exiting the NS, more viral cores lost GFP-CA during the 10-min observation period (18 of 48; ~38%) than in DMSO-treated cells (4 of 35; ~11%) (Fisher's exact test, *P* = 0.006; Supplementary Fig. 2a). However, this is fewer than compared to our earlier studies, where most (86–97%) disassembled following PF74 treatment within a longer observation period (>1 h)[8,52]. This discrepancy likely reflects the shorter imaging

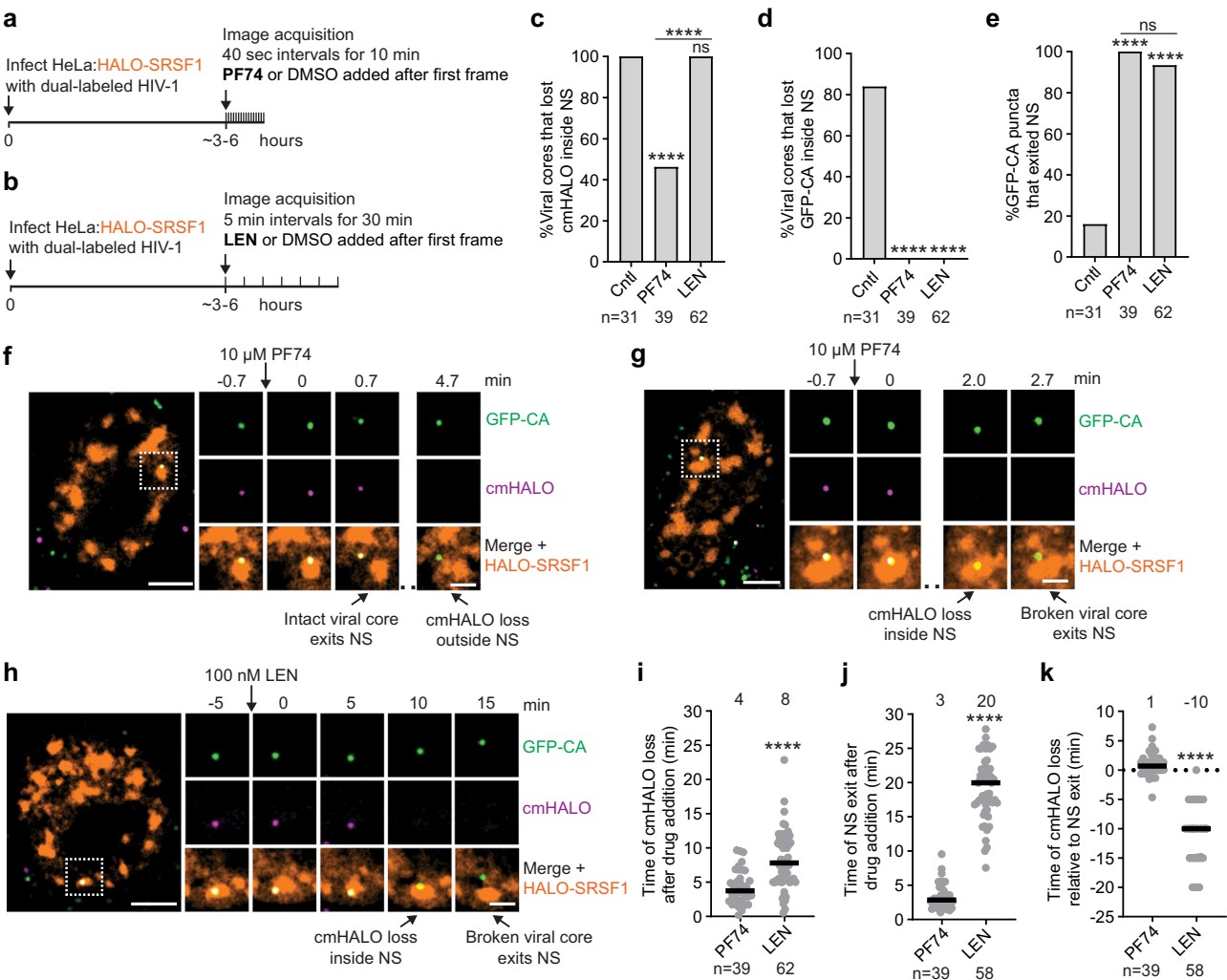

**Fig. 2 | Capsid inhibitors alter HIV-1 uncoating location. a,b** Schematic of live-cell imaging assays to determine effect of PF74 (**a**) or LEN (**b**) on viral cores dual-labeled with cmHALO and GFP-CA and NSs labeled with HALO-SRSF1. **c** Percentage of viral cores that lost cmHALO inside NS. **d** Percentage of viral cores that lost cmHALO and subsequently lost GFP-CA inside NS. **e** Percentage of GFP-CA⁺ puncta that exited NSs preceding or following loss of cmHALO. In (**c**–**e**), statistical significance was determined using two-sided Fisher's exact tests. In the control sample, only viral cores that lost cmHALO and GFP-CA (uncoated) were analyzed (same as in Fig. 1). **f** Representative viral core (1 of 21) that exited NS and subsequently ruptured (lost

the cmHALO signal) following PF74 treatment. **g** Representative viral core (1 of 18) that ruptured inside NS and exited NS following PF74 treatment. **h** Representative viral core (1 of 58) that ruptured inside NS and exited NS following LEN treatment. In (**f**–**h**), scale bars, 5 μm; insets, 2 μm. **i** Time of cmHALO loss relative to PF74 or LEN addition. **j** Time of NS exit relative to PF74 or LEN addition. **k** Time of cmHALO loss relative to the time of NS exit. In (**i**–**k**), lines = median. Statistical significance was determined using two-sided Mann-Whitney U tests. In (**c**–**e**) and (**i**–**k**), number of viral cores analyzed are indicated below graphs. ****$p < 0.0001$, ns (not significant, $p > 0.05$). Source data are provided as a Source Data file.

duration in the current experiments, which was necessary to capture the rapid dynamics of cmHALO loss and NS exit. Quantitative analysis revealed that GFP-CA loss occurred significantly faster in PF74-treated cells (<1 min after cmHALO loss; Supplementary Fig. 2b, c) compared to untreated cells (~8 min; Fig. 1e), indicating that PF74 accelerates capsid lattice disassembly following an initial rupture.

Unlike PF74 treatment and similar to uncoating in untreated cells, all viral cores (100%) lost the cmHALO signal inside NSs following LEN treatment (Fig. 2c; Supplementary Fig. 2d). Similar to PF74 treatment, no GFP-CA loss was observed inside NSs (Fig. 2d) and most (94%) viral cores subsequently exited NSs (Fig. 2e). A representative example of a viral core that lost cmHALO before exiting NS is shown in Fig. 2h and Supplementary Movie 5. Only a small percentage of viral cores lost the GFP-CA signal during the observation period (3%; Supplementary Fig. 2d), consistent with LEN's ability to disrupt the integrity of the core while stabilizing the capsid lattice[48,51–53].

Although both capsid inhibitors induced viral core integrity loss with largely similar kinetics (~4–8 min; Fig. 2i), the timing of NS exit

differed markedly. Viral cores exited NSs ~20 minutes after LEN treatment, whereas PF74 triggered NS exit within ~3 min (Fig. 2j). Since CPSF6 has been implicated in viral core localization to NSs[34,35], these results suggest that PF74 can displace CPSF6 from the viral cores more efficiently than LEN. Moreover, LEN-induced viral core rupture preceded NS exit by ~10 min, while PF74-induced rupture occurred ~1 min after NS exit (Fig. 2k).

The role of CPSF6 in the localization of HIV-1 cores to NSs was examined using a HeLa-derived cell line in which endogenous CPSF6 was knocked out (HeLa:CKO) and an exogenous SNAP-tagged CPSF6 was expressed (HeLa:CKO + SNAP-CPSF6) (Supplementary Fig. 3a). Consistent with our previous studies showing viral cores bearing capsid mutations that disrupt CPSF6 binding do not enter the nucleus[8], the nuclear import of GFP-CA-labeled HIV-1 cores was reduced to background levels in HeLa:CKO (Supplementary Fig. 3b). Expression of SNAP-CPSF6 in HeLa:CKO cells partially rescued nuclear import, reaching ~50% of the level observed in parental HeLa cells (Supplementary Fig. 3b). Virus infectivity in both HeLa:CKO and

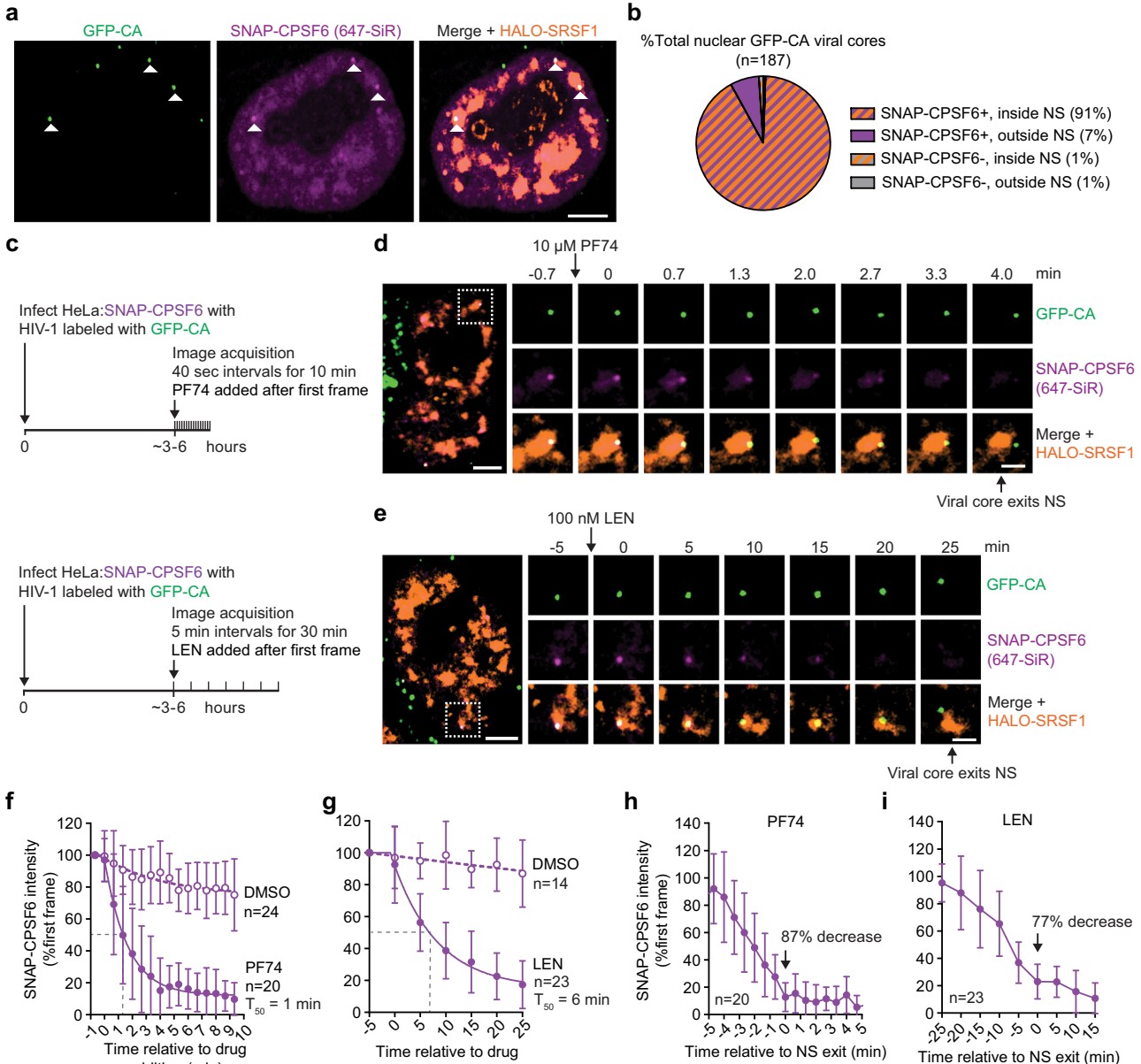

**Fig. 3 | Relocalization of HIV-1 cores from nuclear speckles following displacement of SNAP-CPSF6 by PF74 or LEN. a** Representative image of a HeLa:CKO cell expressing SNAP-tagged CPSF6 (647-SiR) and infected with GFP-CA-labeled HIV-1 (1 of 50). Nuclear HIV-1 cores and associated SNAP-CPSF6 signals are indicated by white triangles. Scale bar, 5 μm. **b** Percentage of nuclear GFP-CA–labeled viral cores that are cmHALO+ or cmHALO− and their localization to NSs based on colocalization with HALO-SRSF1 at 6 hours post-infection (187 viral cores analyzed). **c** Schematics of live-cell imaging assays to determine effect of PF74 (top) or LEN (bottom) on SNAP-CPSF6 association of nuclear GFP-CA-labeled viral cores and their localization to NSs. **d,e** Representative examples showing SNAP-CPSF6 dissociation from a viral core, which subsequently exited an NS following PF74 (1 of 20) (**d**) or LEN (1 of 23) (**e**) treatment. In (**d,e**), scale bars, 5 μm; insets, 2 μm. **f,g** Fluorescence intensity of SNAP-CPSF6 after PF74 treatment (**f**; solid circles) or LEN treatment (**g**; solid circles) compared to DMSO control (open circles); intensities are normalized to the frame prior to drug addition. Curve fitting was applied to determine the time point at which 50% of the SNAP-CPSF6 was lost. **h,i** SNAP-CPSF6 intensity relative to the time of viral core exit from NSs following PF74 (**h**) or LEN (**i**) treatment. In (**f–i**), error bars represent standard deviation. Source data are provided as a Source Data file.

HeLa:CKO + SNAP-CPSF6 remained within ~2-fold of parental cells (Supplementary Fig. 3c). We then modified the SNAP-CPSF6-expressing cells to also stably express HALO-SRSF1. Following infection of these cells with GFP-CA-labeled virus, ~91% nuclear viral cores colocalized with punctate SNAP-CPSF6 signals and localized to HALO-SRSF1-labeled NSs (Fig. 3a, b). We used live-cell imaging to assess the effect of PF74 or LEN addition after viral cores had entered the nucleus and localized to NSs (Fig. 3c). PF74 and LEN treatment led to rapid dissociation of SNAP-CPSF6 from viral cores, followed by the cores leaving the NSs. Representative examples are shown in Fig. 3d and

Supplementary Movie 6, and Fig. 3e and Supplementary Movie 7, respectively. Quantitative analysis showed that ~50% of the capsid-associated SNAP-CPSF6 signal was lost within ~1 min of PF74 treatment (Fig. 3f) and within ~6 minutes of LEN treatment (Fig. 3g). By the time viral cores exited NSs, the SNAP-CPSF6 signal had decreased by ~87% and ~77% following treatment with PF74 or LEN, respectively (Fig. 3h, i). Together, these findings indicate that both PF74 and LEN rapidly displace CPSF6 from HIV-1 cores, and that dissociation of the majority of CPSF6 (~77–87%) correlates with viral core exit from NSs.

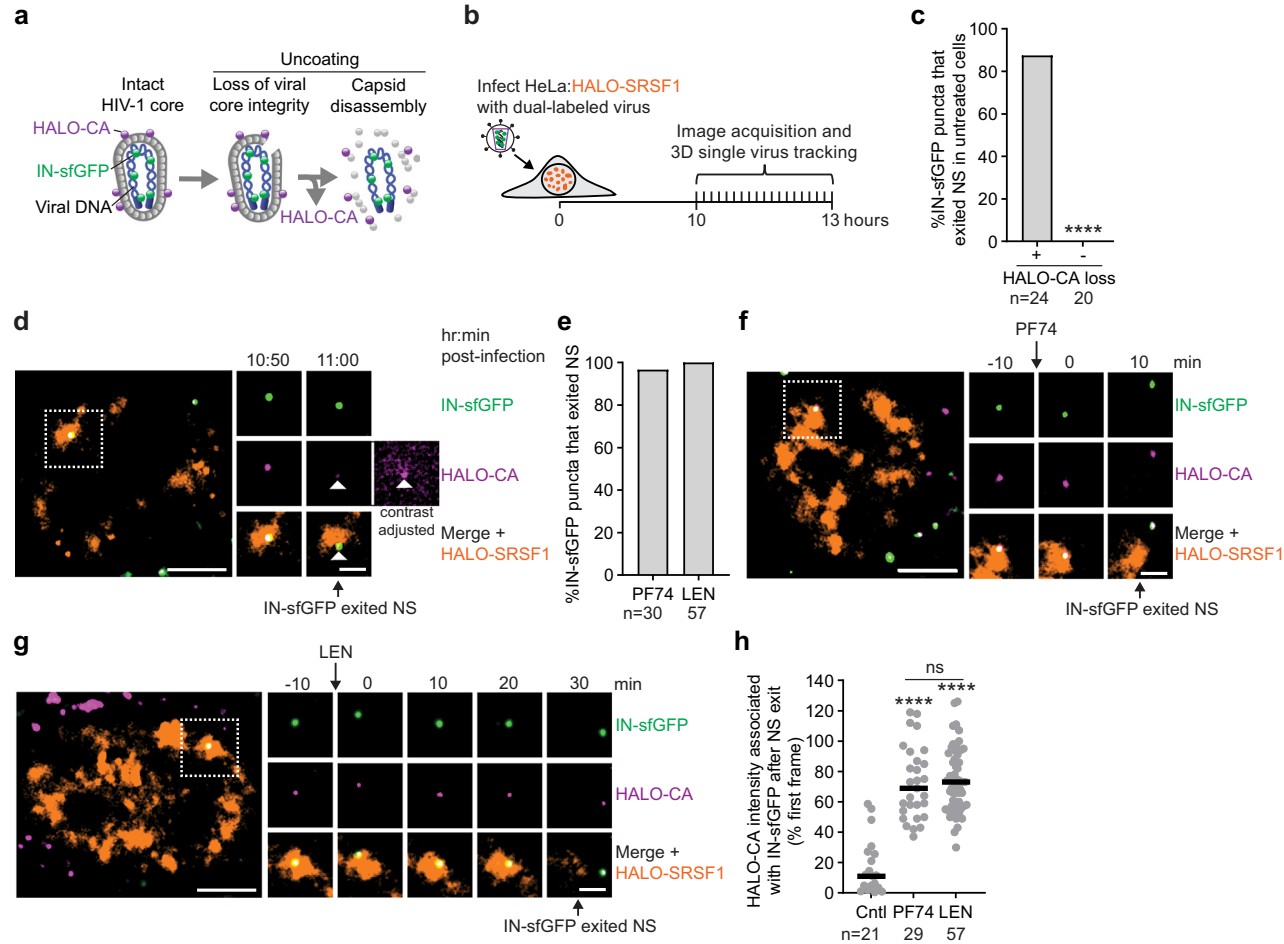

**Fig. 4 | IN-sfGFP complexes exit nuclear speckles following capsid disassembly.**
**a** Schematic of an HIV-1 core labeled with HALO-tagged capsid protein conjugated to JF646 (HALO-CA; purple spheres) and integrase fused to sfGFP (IN-sfGFP; green spheres). Loss of HALO-CA reflects disassembly of the capsid lattice, while IN-sfGFP remains associated with vDNA. **b** Schematic of live-cell imaging assay to visualize uncoating of HALO-CA-labeled viral cores and IN-sfGFP and localization to NSs. Viral core uncoating events were identified between 10-12 hpi, and IN-sfGFP was tracked for an additional hour. For some experiments PF74 or LEN were added after the first frame. **c** Percentage of IN-sfGFP puncta that exited NSs in the presence or absence of uncoating (defined by HALO-CA loss). Statistical significance was determined using a two-sided Fisher's exact test. **d** Representative IN-sfGFP

complex (1 of 21) exiting an NS following substantial loss of HALO-CA in untreated control cells. The contrast-adjusted image shows low HALO-CA signal at the time of NS exit, indicated by the white arrow. **e** Percentage of IN-sfGFP puncta that exited NSs following PF74 or LEN treatment. **f,g** Representative examples showing an IN-sfGFP complex exiting an NS following treatment with PF74 (1 of 29) (**f**) or LEN (1 of 57) (**g**). In d,**f**,**g**, scale bars, 5 μm; insets, 2 μm. **h** HALO-CA intensity associated with IN-sfGFP signal at the time of NS exit. Lines = median. Statistical significance was determined using two-sided Mann-Whitney U tests. In **d**,**f**,**g** Scale bars, 5 μm; 2 μm (insets). In c,**e**,**h**, number of viral cores analyzed are indicated below graphs.
****$p < 0.0001$, ns (not significant, $p > 0.05$). Source data are provided as a Source Data file.

Uncoating and the subsequent vDNA release are critical steps in HIV-1 infection. To track vDNA post-uncoating, we generated virus particles labeled with a HALO-CA fusion protein (analogous to GFP-CA) and HIV-1 integrase fused to superfolder GFP (IN-sfGFP) (Fig. 4a). We hypothesized that the IN-sfGFP will remain associated with the viral PICs and the associated vDNA after uncoating. We identified 24 dual-labeled nuclear viral cores that underwent uncoating, defined by the loss of HALO-CA signal, between 10 and 12 hpi, and then tracked the IN-sfGFP signal for up to an additional hour (Fig. 4b). In most cases (~88%), IN-sfGFP puncta exited NSs following uncoating, whereas in the absence of uncoating, IN-sfGFP puncta, together with the associated HALO-CA signal, remained confined within NSs (Fig. 4c). A representative example of an IN-sfGFP complex exiting an NS upon substantial, but not complete, loss of HALO-CA is shown in Fig. 4d and Supplementary Movie 8. Upon PF74 or LEN treatment, >97% of IN-sfGFP puncta exited NSs (Fig. 4e). Representative examples are shown in Fig. 4f and Supplementary Movie 9, and Fig. 4g and Supplementary Movie 10, respectively. Quantification of HALO-CA intensity revealed

an ~89% reduction at the time of NS exit following uncoating in untreated cells (Fig. 4h), suggesting that extensive capsid disassembly precedes vDNA exit from NSs. By contrast, HALO-CA levels after NS exit were substantially higher following PF74 and LEN treatment (Fig. 4h), consistent with inhibitor-mediated CPSF6 displacement and viral core relocalization from NSs (Fig. 3). Together, these results suggest that NS exit follows the loss of CPSF6, either through capsid disassembly in untreated cells or displacement of capsid-associated CPSF6 in PF74- or LEN-treated cells.

We previously showed that both PF74 and LEN can trigger uncoating of nuclear viral cores and release vDNA capable of integrating into the host genome[54]; however, the chromosomal and spatial location of vDNA integration following inhibitor treatment was not determined. To investigate the spatial relationship between the integration site resulting from capsid inhibitor-induced uncoating and NSs, we used an approach in which transcriptionally active proviruses are visualized by specific recognition of Bgl RNA stem loops in the HIV-1 RNA with Bgl-mCherry[58]. We recently showed that viral cores

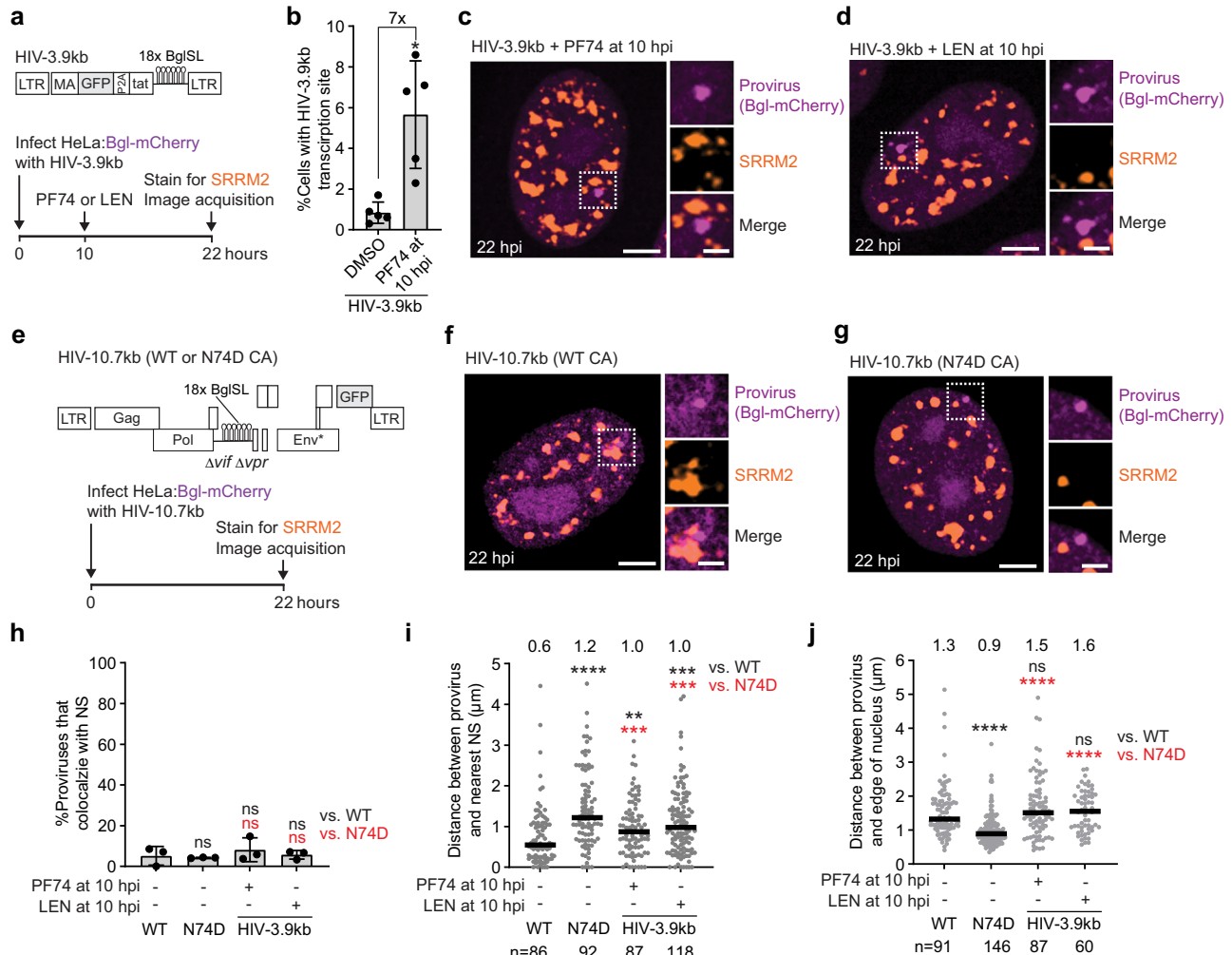

**Fig. 5 | Capsid-inhibitor induced uncoating shifts provirus location away from nuclear speckles. a** Schematic of the short vector (HIV-3.9 kb) containing 18 copies of Bgl stemloop (top) and assay to visualize proviruses after capsid inhibitor-induced uncoating (bottom). **b** Percentage of cells with HIV-3.9 kb transcription sites 22 h post-infection (*n* = 5 independent experiments). Error bars represent standard deviation. Statistical significance was determined using a two-sided Welch's t-test. **c,d** Representative transcriptionally active provirus derived from PF74-induced uncoating (1 of 87) (**c**) or LEN-induced uncoating (1 of 118) (**d**) of viral cores containing HIV-3.9 kb genome. **e,** Schematic of the full-length vector (HIV-10.7 kb) containing 18 copies of Bgl stem loop encoding either WT CA or the CPSF6-binding mutant N74D CA (top) and assay to visualize proviruses after uncoating in untreated cells (bottom). **f,g** Representative provirus derived from uncoating of viral core bearing WT CA (1 of 86) (**f**) or N74D CA (1 of 92) (**g**) and containing HIV-10.7 kb in untreated cells. In **c,d,f,g**, scale bars, 5 μm; insets, 2 μm. **h** Percentage of proviruses that colocalize with NSs (*n* = 3 independent experiments). Error bars represent standard deviation. Statistical significance was determined using two-sided Welch's t-tests. **i** Distance between each provirus and edge of the nearest NS. **j** Distance between provirus and edge of nucleus. In **i,j**, statistical significance was determined using two-sided Mann-Whitney U tests. Number of transcription sites analyzed are indicated below graphs. Lines = median. ****$p < 0.0001$, ***$p < 0.001$, **$p < 0.01$, *$p < 0.05$, ns, not significant ($p > 0.05$). Source data are provided as a Source Data file.

containing short genomes (<3.5 kb) do not uncoat efficiently but treatment with capsid inhibitors induces uncoating followed by vDNA integration[54]. We generated a short viral vector containing multiple copies of the Bgl stem loop (HIV-3.9 kb) and visualized HIV-3.9 kb transcription sites after core disruption with capsid inhibitors (Fig. 5a). Infection of cells expressing Bgl-mCherry with HIV-3.9 kb, followed by viral core disruption with PF74 10 h after infection resulted in a 7-fold increase in the number of cells containing a transcription site compared to the DMSO control (Fig. 5b), consistent with our previous observation that increased PF74-induced uncoating enhances the integration efficiency of short vectors[54].

NSs were visualized by immunofluorescence staining using an antibody targeting the NS marker serine/arginine repetitive matrix 2 (SRRM2) and similar experiments were performed. Proviruses resulting from PF74- and LEN-induced uncoating were observed (examples shown in Fig. 5c, d, respectively). As controls, HIV-1 transcription sites were visualized after infection of untreated HeLa:Bgl-mCherry cells with a full-length HIV-1 vector containing multiple copies of the Bgl stem loops (HIV-10.7 kb) bearing WT CA or CPSF6-binding mutant N74D CA (Fig. 5e; examples of WT and N74D proviruses shown in Fig. 5f, g, respectively). We quantified provirus colocalization with NS masks generated using the SRRM2 signal (Supplementary Fig. 4) and found that, consistent with NSs residing in the interchromatin space, only a small fraction of proviruses (~6%) colocalized with NSs (Fig. 5h). We then measured the distance between each provirus and nearest NS and determined that WT proviruses were located ~0.6 μm from NSs (Fig. 5i), consistent with our earlier observations[24]. In contrast, N74D proviruses were located ~1.2 μm from NSs (Fig. 5i). Notably, proviruses resulting from capsid inhibitor-induced uncoating were located ~1.0 μm from NSs, approximately 0.4 μm further from NSs than WT proviruses. No significant difference was observed between PF74- and LEN-induced uncoating. Measurements of the distance between each

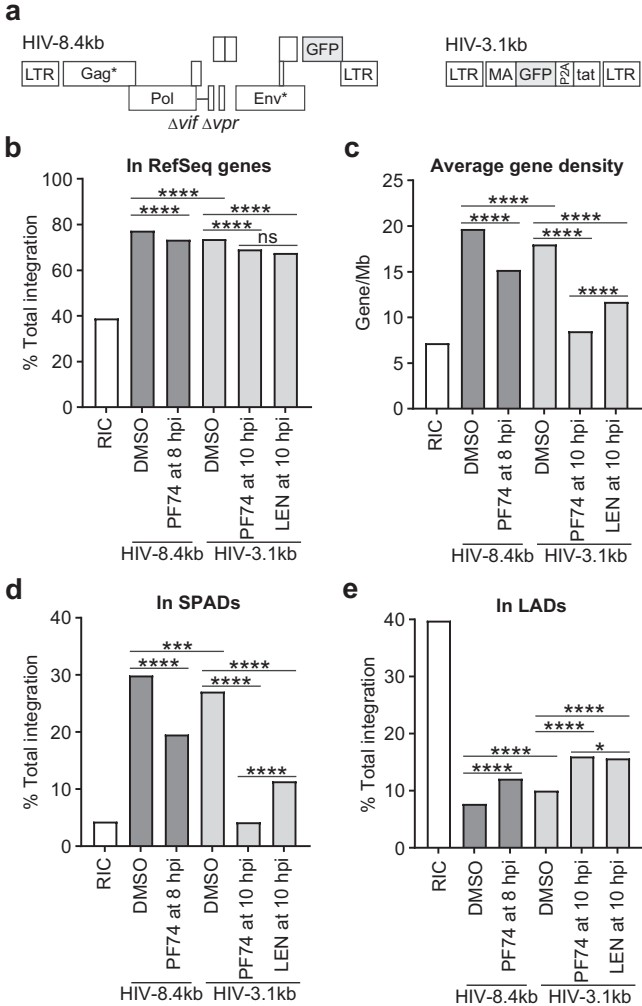

**Fig. 6 | Capsid-inhibitor induced uncoating decreases integration into SPADs.**
**a** Schematic of the near-full-length vector; asterisk indicates a premature stop codon in *gag* (HIV-8.4 kb; left) and the short vector (HIV-3.1 kb; right). An HIV-1 helper plasmid was used for virus production. **b**–**e** HeLa cells were infected with either HIV-8.4 kb (dark shaded bars) or HIV-3.1 kb (light shaded bars), with or without addition of capsid inhibitors at the indicated times, and integration sites were determined. **b** Percentage of total integrations into genes annotated in the NCBI Reference Sequence (RefSeq) database. Statistical significance was determined using two-sided Fisher's exact tests. **c** Average gene density (genes/Mb) surrounding integration site. Statistical significance was determined using two-sided Mann-Whitney U tests. **d** Percentage of total integrations within SPADs. Statistical significance was determined using two-sided Fisher's exact tests. **e** Percentage of total integrations within LADs. Statistical significance was determined using two-sided Fisher's exact tests. RIC, random integration control. Additional statistical comparisons are shown in Supplemental Tables 2–5. ****$p < 0.0001$, ***$p < 0.001$, *$p < 0.05$, ns, not significant ($p > 0.05$). Source data are provided as a Source Data file.

provirus and the nuclear edge (Supplementary Fig. 4) showed that, as expected[8,24], WT proviruses were located ~1.3 μm from the nuclear periphery, whereas N74D proviruses were located significantly closer (~0.9 μm) (Fig. 5j). Proviruses resulting from PF74- or LEN-induced uncoating were located a similar distance from nuclear periphery as WT proviruses (~1.5 μm and ~1.6 μm, respectively; $P > 0.05$). Overall, these findings indicate that proviruses formed following capsid inhibitor-induced uncoating are located farther from NSs, consistent with our observations that these inhibitors cause rapid relocalization of viral cores from NSs. Moreover, the differing locations of viral core rupture and cmHALO loss after PF74 treatment (primarily after NS exit)

and LEN treatment (within NSs), did not result in significant differences in the locations of the proviruses. This suggests that both inhibitors lead to vDNA release outside of NSs, with LEN-induced release of vDNA occurring long after loss of viral core integrity.

We mapped integration sites resulting from two distinct uncoating mechanisms: natural, reverse transcription-dependent uncoating that primarily occurs within NSs, and capsid inhibitor-induced uncoating that mainly occurs outside of NSs. Compared to the random integration control (RIC), an in silico dataset simulation of random integration across the human genome, a near full-length vector (HIV-8.4 kb; Fig. 6a) in untreated cells showed increased integration within genes (77.4% vs. 38.9%; Fig. 6b), gene-rich regions (Avg. gene density ± 500 kb = 19.7 vs. 7.2; Fig. 6c), and SPADs (29.9% vs. 4.3%; Fig. 6d), along with reduced integration within LADs (7.7% vs. 39.8%; Fig. 6e; Supplementary Table 1; Statistical comparisons are shown in Supplementary Tables 2-5). These patterns are consistent with prior studies of HIV-1 integration into cell lines and in peripheral blood mononuclear cells (PBMCs)[25,40,59] (Supplementary Table 1), suggesting that factors involved in proper integration site targeting, including the uncoating site, host factor interactions, and chromatin organization, are conserved across different cell types.

The integration site distribution of the short vector HIV-3.1 kb (Fig. 6a) was evaluated following disruption of viral cores by PF74 or LEN treatment 10 hours post-infection, after many viral cores had entered the nucleus and completed reverse transcription[54]. While the overall efficiency of integration into genes was largely similar to that of the HIV-8.4 kb vector (69.2% and 67.6% for PF74 and LEN, respectively, vs. 77.4% for the 8.4-kb vector in untreated cells; Fig. 6b and Supplementary Table 1), integration of the short HIV-3.1 kb vector shifted toward genomic regions with lower gene density (8.5 and 11.7 for PF74 and LEN, respectively, vs. 19.7; Fig. 6c), showed a substantial decrease in integration into SPADs (4.2% and 11.4% for PF74 and LEN, respectively, vs. 29.9%; Fig. 6d), and a modest increase in integration into LADs (16.0% and 15.7% for PF74 and LEN, respectively, vs. 7.7%; Fig. 6e). LEN treatment resulted in a modestly higher frequency of SPAD integration than PF74 treatment (11.4% vs. 4.2%; $P < 1 \times 10^{-15}$), although the underlying mechanism for this difference remains unclear. Integration of the short vector following treatment with 10 nM LEN, a concentration maintained in plasma for >6 months after a single subcutaneous dose of LEN[60] and sufficient to disrupt viral cores[52,54] indicated that despite the expected slower kinetics of viral core integrity loss[52], a similar proportion of integrations occurred in LADs compared to that observed with 100 nM LEN treatment ($P = 0.07$; Supplementary Tables 1 and 5).

Despite inefficient uncoating of viral cores containing short genomes, integration can still occur, though with reduced efficiency and a significant delay[54]. In untreated cells, the short vector DNA integrated into SPADs with a similar frequency to that by the near full-length vector (Fig. 6b–e; Supplementary Table 1; $P = 0.07$ compared to NL4-3 PBMC; Supplementary Table 4), indicating that the altered integration pattern observed following capsid inhibitor treatment was not due to vector length differences. To assess whether integration of the near full-length vector could be altered by capsid inhibitor–induced uncoating, we treated cells infected with the HIV-8.4 kb vector with PF74 at 8 hours post-infection, a time when many viral cores have completed reverse transcription, uncoated and integrated their DNA into the host genome, while many other viral cores have completed reverse transcription but have not yet uncoated[8,54]. Integration efficiency into genes remained comparable to other samples (Fig. 6b); however, there was a decrease in integration within gene-rich regions (15.2 vs. 19.7%) and SPADs (19.6% vs. 29.9%) (Fig. 6c, d). These modest reductions likely reflect a mixture of integration events resulting from natural, reverse transcription-dependent uncoating inside NSs occurring prior to PF74 addition and PF74-induced uncoating outside NSs.

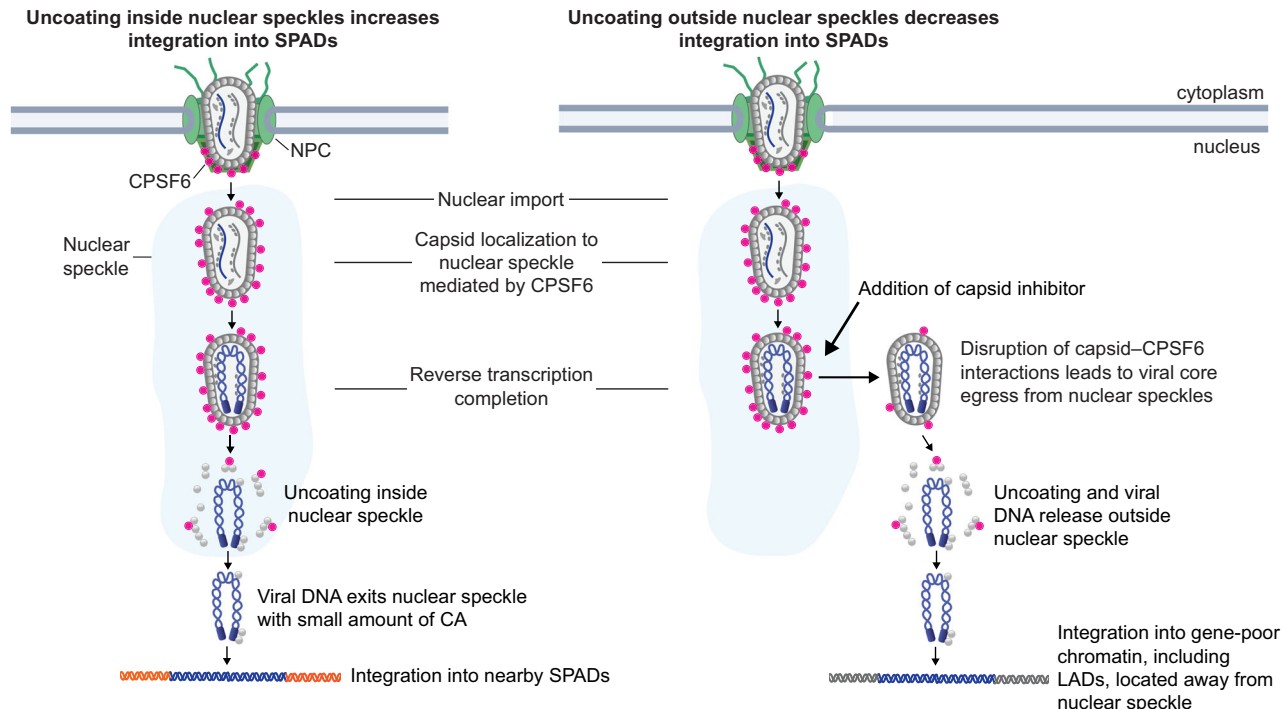

**Fig. 7 | Model depicting spatial relationship between the HIV-1 uncoating site and integration site.** Left, Intact HIV-1 core enters the nucleus through a nuclear pore complex (NPC) and localizes to a nuclear speckle (NS) via binding to CPSF6 (red spheres). Upon completion of reverse transcription, loss of viral core integrity and capsid disassembly occur within the NS; the released viral DNA (blue helix) exits the NS and integrates into nearby speckle-associated domains (SPADs; red helices), the preferred target of HIV-1 integration. Right, Treatment with capsid inhibitors PF74 or LEN after nuclear entry and viral core localization to a NS displaces CPSF6, causing the viral core to exit the NS, followed by uncoating and viral DNA release outside the NS. This shifts integration from SPADs to gene-poor chromatin, including lamina-associated domains (LADs; gray helices), which are located away from both NSs and the nuclear envelope.

## Discussion

HIV-1 uncoating is a critical early step in infection that precedes the vDNA integration into host cell chromosomes. Here, we demonstrate that loss of viral core integrity and subsequent uncoating occur within NSs (model presented in Fig. 7). CPSF6 associates with viral cores docked at the nuclear envelope and facilitates their nuclear import and localization to NSs. In untreated cells, reverse transcription is completed and viral cores uncoat inside the NSs. The viral DNA associated with a small amount of CA exits the NSs and integrates into nearby SPADs to form a transcriptionally active provirus. However, when infected cells were treated with PF74 or LEN after viral cores had entered the nucleus and localized to NSs, CPSF6 was displaced from the viral cores, the viral cores subsequently exited the NSs, and uncoating occurred outside of NSs. Uncoating outside of the NSs resulted in decreased integration into SPADs and increased integration into gene-poor chromatin, including LADs, located away from both NSs and the nuclear envelope. The correlation between CPSF6 displacement and exit of the viral core from NSs suggests that CPSF6 binding is required to retain cores within NSs, which promotes integration into SPADs/gene-dense regions. Taken together, these findings indicate that in cells treated with capsid inhibitors, the uncoating site shifts from within NSs to outside of NSs, which changes the integration site distribution from chromatin near the NSs (SPADs) to gene-poor regions of chromatin away from NSs, including LADs.

Although both PF74 and LEN induced rapid loss of viral core integrity (~4–8 min), the kinetics of CPSF6 displacement and viral core relocalization from NSs were markedly different. CPSF6 dissociation and viral core exit from NSs occurred significantly faster with PF74 than with LEN, resulting in the rapid relocalization of many intact viral cores following PF74 treatment. In contrast, LEN induced rapid viral core breakage but delayed CPSF6 dissociation, leading to slower relocalization of broken viral cores from NSs. The basis for these kinetic differences in the loss of capsid-associated CPSF6 remains unclear but may involve greater accessibility or higher occupancy of FG-binding pockets on the capsid surface by PF74 compared to LEN. Despite their opposing effects on capsid stability—PF74 promoting capsid disassembly and LEN stabilizing broken capsids—both treatments ultimately led to vDNA being released outside NSs, as evidenced by similar provirus localization and integration site patterns.

The uncoating process of HIV-1 cores in the nucleus and the timing of vDNA release are poorly understood, largely due to a lack of robust assays capable of capturing these rare and transient events. Our observation that nuclear capsids disassemble following the initial loss of viral core integrity is consistent with a recent study that used a fluid-phase yellow fluorescent protein (YFP) marker to assess viral core integrity and employed direct labeling of CA (done using an expanded genetic code) to monitor capsid disassembly[13]. We also found that IN-sfGFP-labeled vDNA is released from NSs only after substantial capsid disassembly. The continued association of IN-sfGFP with the PICs enabled us to determine that low levels of HALO-CA signal remained transiently associated with the IN-sfGFP signal shortly after the PICs exited NSs. The mechanism by which small amounts of CA remain associated with the PIC, and whether this residual CA plays a functional role in PIC formation and/or in vDNA integration, remains unclear. A recent report proposed that vDNA separates from cone-shaped capsids with breaks within the nucleus[15]. The apparent discrepancy with our conclusions may be explained by the strong CA signal detected in the Müller et al. experiments, suggesting the presence of multiple capsids within a single focus, and disassembly of one capsid being masked by overlapping signals from intact capsids. Collectively, our data supports a model in which vDNA is released from capsids within NSs during disassembly. The vDNA subsequently exits the NSs (as part of a PIC), traveling a short distance to reach its integration site.

PF74 and LEN potently inhibit nuclear import, making it unlikely that HIV-1 capsids in the nucleus would encounter the high inhibitor concentrations used in our experiments. Prior studies have examined integration site distributions following PF74 or LEN treatment at the time of infection and reported significantly altered integration patterns compared to untreated controls, including reduced integration within well-annotated human genes from the National Center for Biotechnology Information Reference Sequence database (RefSeq) and gene-dense regions, and increased integration into LADs[48,61,62]. Although these studies did not assess uncoating, the inhibitor concentrations used are expected to potently block nuclear import[45,48,63,64]. Thus, the observed changes in integration sites may reflect uncoating at the nuclear periphery, followed by integration into heterochromatic, gene-poor regions near the nuclear envelope—similar to the pattern observed with CPSF6-binding-defective capsid mutants whose cores uncoat while docked at the NPCs. Interestingly, post-nuclear entry treatment with PF74 or LEN also led to increased integration into LADs that were located away from NSs and the nuclear periphery. This is consistent with a prior report showing that ~70% of LADs are located away from the nuclear periphery due to stochastic shuffling of chromosomes upon mitosis[65]. In contrast to earlier studies, our data directly demonstrates that shifting the site of nuclear uncoating through post-nuclear entry treatment with capsid inhibitors is sufficient to redirect integration away from SPADs and toward chromatin regions located farther from NSs. These findings support a model in which the uncoating site plays a pivotal role in determining integration site selection.

Collectively, our data show that HIV-1 preferentially integrates into chromatin regions located near the site of uncoating. Based on the behavior of IN-sfGFP labeled viral complexes and localization of proviruses, we infer that the vDNA (as part of the PIC) is released within NSs during capsid disassembly, subsequently exits NSs, and then travels a short distance to the eventual integration site in nearby SPADs. Several key questions remain unresolved, including when and where intasome assembly occurs, when the intasome first encounters LEDGF/p75, and what mechanisms guide the intasome to its chromatin target. It also remains unclear whether the small amount of capsid that remains associated with the vDNA upon NS exit is a critical component of the PIC, as previously proposed[50]. Addressing these questions will be essential for elucidating the molecular determinants of integration site selection following uncoating, with potential implications for the regulation of viral gene expression and for the establishment and reactivation of the latent viral reservoir.

## Methods

### Cell lines and reagents
HeLa [American Type Culture Collection (ATCC CCL-2)], HeLa derived cell lines (described below), TZM-bl cells[66], human embryonic kidney (HEK) 293 T cells (ATCC CRL-3216), and HEK-derived cell line GP2-293 (Takara Bio) were maintained in Dulbecco's modified Eagle's medium (DMEM) supplemented with 10% fetal calf serum (Hyclone), 100 Units/ml penicillin, and 100 μg/ml streptomycin (Gibco). The HeLa-Bgl cell line expresses truncated bacterial protein BglG that is fused to mCherry at the C-terminus and contains a nuclear localization signal (Bgl-mCherry) from a doxycycline-inducible promoter[8]. HeLa-derived cell lines were cultured in medium containing the appropriate antibiotics (2 μg/ml puromycin and/or 100 μg/ml hygromycin). All cells were maintained in a humidified incubator at 37 °C with 5% carbon dioxide. PF74 (Sigma) and LEN (GS-6207; MedChemExpress) were used at a final concentration of 10 μM and 100 nM, respectively, unless indicated otherwise.

### Plasmids
pHmNG is a full-length HIV-1-based vector that contains an *mNeon-Green* reporter gene (mNG) in place of *nef* and does not express Env. pHmNG was generated by replacing the GFP reporter gene in pHGFP[54]

with mNG. In pHGFP-GFPCA[8] a GFP coding sequence is located between the matrix (MA) and capsid (CA) coding sequences. A mutation of the protease cleavage site (Y132I) between the GFP and CA coding sequences results in expression of a GFP-CA fusion protein following proteolytic processing of the Gag polyprotein; an additional GFP reporter is in place of *nef*; the vector does not express Env. pHGFP-HALOCA is similar to pHGFP-GFPCA, except that a HaloTag coding sequence (HALO; Promega) is located between the MA and CA coding sequences. pHmNG-iHALO contains a HALO coding sequence between the MA and CA coding sequences with functional protease cleavage sites, resulting in expression of free HALO and CA proteins after Gag processing. To generate pHmNG-iHALO, the GFP in HIV Gag-iGFP ΔEnv[67] (NIH AIDS Reagent Program, Division of AIDS, NIAID, NIH, from Dr. Benjamin Chen; Cat#12455) was replaced with HALO and then a portion of gag containing HALO was transferred to pHmNG, using standard molecular biology techniques[9,13,56,67]. HIV-8.4 kb is an HIV-1-based vector that contains a frameshift mutation in CA leading to a premature stop codon in *gag*, contains a GFP reporter in place of *nef*, and does not express Vpr, Vif, or Env[54]. HIV-3.1 kb is a short vector that expresses a MA-GFP fusion protein and codon-optimized Tat, separated by an in-frame self-cleaving P2A peptide[54]. HIV-3.9 kb is similar to HIV-3.1 kb, except that it includes a cassette containing 18 copies of Bgl stem loop, each separated by short (~10 nucleotide) randomized linkers, located downstream of the *GFP* coding sequence. The HIV-1-based vector pHGFP-BglSL[8] (referred to as HIV-10.7 kb in this study) contains a cassette containing 18 copies of Bgl stem loops separated by short, randomized linkers inserted in the Vpr-Vif region and also contains a GFP reporter in place of *nef*. This vector does not express Vif, Vpr, or Env. HIV-10.7 kb(WT) expresses wild-type CA, while HIV-10.7 kb(N74D) expresses a CPSF6-binding-defective CA mutant (N74D)[8]. pcHELP is an HIV-1 helper construct that lacks a packaging signal and primer-binding site and expresses all the viral proteins except Nef and Env[68]. pLR2-Vpr-IN-sfGFP is a vector that expresses a Vpr-IN-sfGFP fusion protein, in which the Vpr and IN-sfGFP components are separated by an HIV-1 protease cleavage site. This construct, inspired by Francis et al.[69], was generated by PCR amplification and Gibson cloning of the sfGFP coding sequence into pLR2-Vpr-IN[70]. pHCMV-G expresses the G glycoprotein of vesicular stomatitis virus (VSV-G)[71]. All plasmids were verified by sequencing (Psomagen).

pMLV-HALO-SRSF1-IRES-Hygro is a Moloney murine leukemia virus (MLV)-based retroviral vector that expresses HALO-tagged serine/arginine splicing factor 1 (SRSF1) and hygromycin resistance gene (Hygro) using an internal ribosomal entry site (IRES), and was constructed by replacing the Tat-P2A-Rev coding sequence in pMLV-Tat-P2A-Rev-IRES-Hygro[8] with gBlocks (Integrated DNA Technologies; IDT) containing HALO and SRSF1 coding sequences using NEBuilder HiFi DNA Assembly Master Mix (New England Biolabs). pMLV-SNAP-CPSF6-IRES-Hygro is a retroviral vector that expresses SNAP-tagged CPSF6 and hygromycin. pMLV-SNAP-CPSF6-IRES-Hygro was constructed by replacing the HALO-SRSF1 coding sequence in pMLV-HALO-SRSF1-IRES-Hygro with SNAP-CPSF6 coding sequence using three overlapping gBlocks and NEBuilder HiFi DNA Assembly Master Mix. pLVX-EF1-HALO-SRSF1-IRES-HSA is a bicistronic lentiviral vector that contains a partial deletion in the U3 of the 3' long terminal repeat and expresses a HALO-tagged SRSF1 fusion protein and heat stable antigen (HSA) surface marker under the control of the human EF1 promoter. pLVX-EF1-HALO-SRSF1-IRES-HSA was constructed by replacing the mRuby-LaminB-P2A-Puro coding sequence in pLVXsin-EF1-mRuby-LaminB-P2A-Puro[54] with a coding sequence containing HALO-SRSF1-IRES-HSA (IDT) and NEBuilder HiFi DNA Assembly Master Mix. Cassettes encoding guide RNAs targeting the 5'-ATAGA-CATTTACGCGGATGT-3' or 5'-ATATTGGAAATCTAACATGG-3' sequences (IDT) located near the 5' end of CPSF6 were cloned into LentiCRISPRv2, a vector that expresses Cas9 and puromycin resistance genes[72] (Addgene plasmid #52961).

## Virus production and infection

Virions labeled with GFP-CA and cmHALO were prepared by polyethylenimine transfection (PEI; Polysciences) of 293 T cells ($3.5 \times 10^6$ cells seeded in 100-mm cell culture dish one day prior) with the HIV-1-based vectors pHmNG (4 μg), pHGFP-GFPCA (1 μg), and pHmNG-iHALO (5 μg), along with pHCMV-G (VSV-G expression plasmid; 2 μg). Virions labeled with GFP-CA were prepared by PEI transfection of 293 T cells with pHmNG (9 μg) and pHGFP-GFPCA (1 μg), along with VSV-G (2 μg). Virions labeled with HALO-CA and IN-sfGFP were prepared by PEI transfection of 293 T cells with pHmNG (9 μg) and pHGFP-HALOCA (1 μg) along with pLR2P-Vpr-IN-sfGFP (5 ug) and VSV-G (2 μg). Unlabeled virions were prepared by PEI transfection of 293 T cells with pHmNG (10 μg) and VSV-G (2 μg). Retroviral particles containing a bicistronic vector that expresses HALO-SRSF1 and hygromycin were prepared by PEI transfection of GP2-293 cells, a HEK-derived packaging cell line that expresses the MLV Gag and Pol proteins (Takara Bio), with pMLV-HALO-SRSF1-IRES-Hygro (10 μg) and VSV-G (2 μg). Retroviral particles containing a bicistronic vector that expresses SNAP-CPSF6 and hygromycin were prepared by PEI transfection of GP2-293 cells with pMLV-SNAP-CPSF6-IRES-Hygro (10 μg) and VSV-G (2 μg). Lentiviral particles containing a bicistronic vector that expresses HALO-SRSF1 and HSA surface marker were prepared by PEI transection of 293 T cells with pLVXsin-EF1-HALO-SRSF1-IRES-HSA (10 μg) and VSV-G (2 μg). Three to four hours after transfection, the transfection mixture was replaced with medium containing 20 nM Janelia Farm 646 (JF646) dye (to produce dual-labeled virus) or medium without dye (to produce unlabeled virions or virions labeled only with GFP-CA). Supernatants from the transfected cells were collected 24 hours after transfection, clarified using a 0.45 μm membrane filter, and concentrated by ultracentrifugation ($100,000 \times g$) for 1.5 h at 4 °C through a 20% sucrose cushion (wt/vol) in Dulbecco's phosphate-buffered saline with calcium and magnesium (PBS). The pelleted virus was resuspended in 500 μl medium.

HeLa-derived cell lines were seeded in 48-well plates (for measuring infectivity) or glass bottom μ-slides ($3 \times 10^4$ cells/well; Ibidi) one day prior to infection. Cells were infected with viruses via spinoculation ($1200 \times g$, 1 h) at 15 °C, which permitted virion binding to cell membranes but prevented virion endocytosis[73], in the presence of 10 μg/ml polybrene (Sigma). After spinoculation, the medium was replaced with prewarmed medium to allow internalization of the virus (defined as the 0-hour time point) and incubated at 37 °C. For infectivity measurements, cells were collected 48 hours after infection using TrypLE cell dissociation solution (Gibco), fixed with 2% paraformaldehyde in 1X PBS for 10 min, and the percentage of GFP reporter–expressing cells was determined by flow cytometry (LSRFortessa; BD Biosciences). For some infection experiments, TZM-bl cells were seeded in 96-well plates ($6 \times 10^3$ cells/well) and challenged the next day with p24-normalized virus via spinoculation ($1200 \times g$, 1 h, 15 °C) in the presence of 10 μg/ml polybrene. After spinoculation, the cells were incubated at 37 °C, and luciferase activity was measured 48 h after infection using the Britelite plus reporter gene assay system (Revvity) and the SpectraMax iD3 Microplate Reader (Molecular Devices).

## Generation of HeLa:HALO-SRSF1

HeLa:HALO-SRSF1, a HeLa-derived cell line expressing HALO-tagged SRSF1 was generated by transduction of Hela cells in a 48-well plate ($3 \times 10^4$ cells/well seeded day before) with VSV-G pseudotyped retroviral particles containing a vector that expresses HALO-SRSF1 and hygromycin resistance gene. Transduced cells were selected with 100 μg/ml hygromycin for 2 weeks. Expression of HALO-SRSF1 was determined by confocal microscopy after incubation of the selected cells in medium containing 20 nM JF646 for 30 min, followed by two rinses with medium. Localization of the HALO-SRSF1 fusion protein to NSs was determined by immunofluorescence staining of SRRM2,

another NS marker, using rabbit anti-SRRM2 antibody (PA5-66827; Invitrogen) followed by donkey anti-rabbit antibody labeled with Alexa Fluor 488 (A21206; Invitrogen).

## Generation of HeLa:CKO, HeLa:CKO + SNAP-CPSF6, and HeLa:CKO + SNAP-CPSF6 + HALO-SRSF1

HeLa:CKO cells were generated by CRISPR-Cas9-mediated knockout of CPSF6 in HeLa cells. HeLa cells were co-infected with two different LentiCRISPRv2 vectors that express guide RNAs targeting the 5′ end of CPSF6. Single cell clones were obtained by limited dilution and selection with 2 μg/ml puromycin. After expansion of single cell clones (>1 month), CPSF6 and tubulin expression were determined by western blot analysis using rabbit anti-CPSF6 antibody (HPA039973; Sigma-Aldrich) and mouse anti-tubulin antibody (T9026; Sigma-Aldrich) followed by goat anti-rabbit (IRDye-800CW; LI-COR) and goat anti-mouse (IRDye-680RD; LI-COR) antibodies. Western blots were imaged using the Odyssey infrared imaging system (LI-COR). Seven single cell clones in which CPSF6 expression was undetectable by western blot analysis and cell division kinetics was similar to that of parental HeLa cells were pooled, generating HeLa:CKO. The HeLa:CKO + SNAP-CPSF6 cell line was generated by transduction of HeLa:CKO cells with VSV-G pseudotyped retroviral vector that expresses SNAP-CPSF6 and a hygromycin resistance gene. Transduced cells were selected with 100 μg/ml hygromycin for 2 weeks. HeLa:CKO + SNAP-CPSF6 + HALO-SRSF1, a cell line in which HeLa:CKO + SNAP-CPSF6 cells express HALO-SRSF1 was generated by transduction of the HeLa:CKO + SNAP-CPSF6 cell line with a VSV-G pseudotyped lentiviral vector that expresses HALO-SRSF1 at a high multiplicity of infection so that many cells express HALO-SRSF1 without additional selection. For movies, nuclei were aligned over time using a custom MATLAB script. Briefly, nuclear masks were generated for each time point using the HALO-SRSF1 signal and aligned to the mask in the first frame. For each time point, this alignment was used to compute a transformation matrix—a mathematical description of how the image must be shifted and rotated to match the reference—which was then applied to all imaging channels. For each time-point, an average projection was generated from the z-slices containing the particle of interest (defined as the z-slice with highest intensity ±1 adjacent z-slice). The aligned images were then imported into ImageJ, smoothed, and a small region containing the example particle was cropped and scaled 3×. Labels and scale bars were added in ImageJ.

## Confocal microscopy and image processing

Confocal images were acquired using a Nikon Eclipse Ti-E microscope equipped with a Yokogawa CSU-W1 spinning disk unit and a Plan-Apochromat 100× N.A. 1.49 oil objective, using 488-nm (GFP), 561-nm (JF549), 594-nm (mCherry), and 640-nm (JF646/BioTracker 650 Red Nuclear Dye/SNAP-Cell 647-SiR) lasers for illumination. Images were captured using a 561-nm long pass dichroic mirror and two ORCA-fusion BT cameras (Hamamatsu). A Tokai Hit microscope stage top incubator was used for all live-cell imaging experiments. Digital images were examined using Nikon Elements or ImageJ[74]. When necessary, channels were aligned using Nikon Elements. For display, a pixel-averaging filter was applied to the images and the contrast was adjusted; unmodified images were used for intensity analyses.

## Characterization of dual-labeled HIV-1 particles

Viral lysates were prepared using Pierce RIPA buffer (ThermoFisher) and subjected to western blot analysis using mouse anti-p24 antibody (obtained through the NIH HIV Reagent Program, Division of AIDS, NIAID, NIH: Monoclonal Anti-Human Immunodeficiency Virus Type 1 (HIV-1) p24 Gag Protein (#24-3), ARP-6458, contributed by Dr. Michael H. Malim), rabbit anti-GFP (ThermoFisher Scientific, A6455), and rabbit anti-HALO (Promega, G9281) followed by IRDye 800CW-labeled goat anti-rabbit secondary antibody (LI-COR, 926-32211) or IRDye 680-

labeled goat anti-mouse secondary antibody (LI-COR, 926-68070). Protein bands were visualized using an Odyssey Infrared Imaging System (LI-COR). Fluorescent virus particles were analyzed by single virion analysis[75]. Virus particles were centrifuged (1200 × g for 30 min) onto glass-bottom 8-well μ-slides (Ibidi) and then imaged using confocal microscopy. GFP and HALO(JF646) signals in each image were detected using Localize[76] and the percentage of GFP signals that colocalized with JF646 signal was determined using a custom MATLAB script (MathWorks).

### Live-cell imaging of uncoating of dual-labeled viral cores in HeLa:HALO-SRSF1 cells

HeLa:HALO-SRSF1 cells were incubated in medium containing 20 nM Halo tag ligand conjugated to Janelia Fluor 561 (Halo-JF549) dye for 30 min followed by two washes with media. Next, cells were infected by spinoculation with VSV-G-pseudotyped HIV-1 virions containing GFP-CA and cmHALO, which were prepared by addition of the JF646 dye during virus production. Time lapse images (z-stack; 12 slices with 0.4 μm step size) were acquired every 10 min between 10- and 12-hours post-infection. For live-cell imaging of PF74-induced uncoating, HeLa:HALO-SRSF1(JF549) cells were challenged with VSV-G-pseudotyped HIV-1 containing GFP-CA and cmHALO(JF646) via spinoculation. Confocal z-stacks (7 slices with 0.4 μm step size) were acquired every 40 sec for 10 min starting between 3 and 6 h after infection, a time when many viral cores entered the nucleus and localized to NSs. After the first frame, 200 μl media containing 2X PF74 (final concentration = 10 μM) or DMSO was added to the well. For live-cell imaging of LEN-induced uncoating, HeLa:HALO-SRSF1(JF549) cells were challenged with VSV-G-pseudotyped HIV-1 containing GFP-CA and cmHALO(JF646) via spinoculation. Confocal z-stacks (12 slices with 0.4 μm step size) were acquired every 5 min for 30 min starting between 3 and 6 h after infection. After the first frame, 200 μl media containing 2X LEN (final concentration = 100 nM) or DMSO was added to the well. The viral cores were tracked and the background subtracted fluorescence intensities of the GFP (GFP-CA), JF646 (cmHALO), and JF549 (NSs) channels at the position of the GFP particle were extracted from the time-lapse images using a custom MATLAB script. In addition, the JF549 fluorescence intensity of the diffuse background at several positions (signals not associated with the nuclear speckle-associated JF549 signal) were extracted. Viral cores were considered localized to NSs if the JF549 intensity at the position of the viral core was ≥1.1-fold higher than the JF549 signal in the diffuse background of the nucleus. The time when 50% of the GFP-CA signal or cmHALO(JF646) signal was lost was determined by fitting the background subtracted intensities with a nonlinear curve fitting (plateau followed by one phase decay; GraphPad Prism 10). The percentage of nuclear GFP-CA–labeled HIV-1 cores that were cmHALO[+] was determined by measuring the cmHALO signal intensity associated with each GFP-CA punctum. A threshold for positivity was defined as the background signal in the same channel plus one standard deviation.

### Live-cell imaging of GFP-CA-labeled viral cores in HeLa:SNAP-CPSF6 + HALO-SRSF1 cells

HeLa:CKO + SNAP-CPSF6 + HALO-SRSF1 cells were challenged with VSV-G-pseudotyped HIV-1 containing GFP-CA via spinoculation. Prior to live-cell imaging, infected cells were incubated with medium containing 100 nM Halo-JF549 and 100 nM SNAP-Cell 647-SiR dye for 60 minutes followed by two washes with medium. For determining the effect of PF74 on viral core association with CPSF6, confocal z-stacks (7 slices with 0.4 μm step size) were acquired every 40 s for 10 min starting between 3 and 6 h after infection. After the first frame, 200 μl media containing 2X PF74 (final concentration = 10 μM) was added to the well. For determining the effect of LEN on viral core association with CPSF6, confocal z-stacks (12 slices with 0.4 μm step size) were acquired every 5 min for 30 min starting between 3 and 6 h after

infection. After the first frame, 200 μl medium containing 2X LEN (final concentration = 100 nM) was added to the well. The GFP-CA-labeled viral cores were tracked and the background subtracted fluorescence intensities of the GFP (GFP-CA), SNAP-Cell 647-SiR (SNAP-CPSF6), and JF549 (NSs) channels at the position of the GFP particle signal were extracted from the time-lapse images using a custom MATLAB script. The time when 50% of the 647-SiR signal was lost was determined by fitting the background subtracted intensities with a nonlinear curve fitting (plateau followed by one phase decay; GraphPad Prism 10). The percentage of nuclear GFP-CA–labeled HIV-1 cores that were SNAP-CPSF6[+] was determined by measuring the SNAP-CPSF6 signal intensity associated with each GFP-CA punctum. A threshold for positivity was defined as the background signal in the same channel plus one standard deviation. NS localization was determined as described above.

### Imaging NSs and transcriptionally active proviruses and image analysis

HeLa:Bgl-mCherry cells were seeded in ibiTreated μ-slides (3 × 10[4] cells/well; Ibidi) one day prior to infection using medium containing 1 μg/ml doxycycline. The next day, cells were challenged with VSV-G-pseudotyped lentiviral particles containing the HIV-3.9 kb vector (short genome containing 18 Bgl stem loops) or HIV-1 full-length genome containing 18 copies of Bgl stem loop (HIV-10.7 kb) and harboring WT CA or N74D CA in medium containing 1 μg/ml doxycycline and 10 μg/ml aphidicolin. For cells infected with HIV-3.9 kb, media containing 10 μM PF74 or 100 nM LEN were added 10 hours after infection. Cells were fixed with 4% paraformaldehyde in PBS approximately 22 hours after infection, a time when many HIV-1 transcription sites can be detected[8], followed by two rinses with PBS. NSs were identified by immunostaining of SRRM2, a splicing factor that is highly enriched in NSs, using mouse anti-SC-35 antibody (S4045; Sigma-Aldrich; note that this antibody primarily targets SRRM2 in immunological assays[28]) followed by goat anti-mouse antibody labeled with Cy5 (A10524; Invitrogen). Cells were stored in PBS and imaged immediately or stored at 4 °C and imaged the next day.

To determine the spatial relationship between the HIV-3.9 kb and HIV-10.7 kb transcription sites and NSs, a confocal z-slice was acquired at the focal plane where the transcription site was best resolved. Colocalization with NSs and the distance between each transcription site and the nearest NS were assessed using a previously described custom MATLAB script[24]. Briefly, the x and y coordinates of each transcription site were manually identified. NS masks were generated based on SRRM2 immunostaining, and each transcription site's overlap with the NS mask and its distance to the edge of the nearest NS were calculated. To assess the spatial relationship between the HIV-3.9 kb and HIV-10.7 kb transcription sites and the nuclear envelope, confocal z-stacks (46 z-slices with 0.2 μm step size) were acquired of infected cells containing a transcription site. The distance from each transcription site and the nuclear boundary was determined using a previously described custom MATLAB script[24]. Briefly, the x, y, and z coordinates of each transcription site were manually identified and further refined using three-dimensional Gaussian fitting. The 3D nuclear boundary was defined using the diffuse Bgl-mCherry signal, which is predominantly localized inside the nucleus due to the presence of a nuclear localization signal. We previously showed that the sharp reduction in diffuse Bgl-mCherry signal colocalizes with nuclear pore proteins[8], indicating that the edge of the Bgl-mCherry signal can be used to approximate the nuclear envelope location. The shortest 3D distance from each transcription site to the nuclear boundary was calculated based on the point of abrupt drop in the Bgl-mCherry signal.

### Quantification of nuclear import efficiency

To quantify nuclear import of HIV-1, HeLa-derived cell lines were seeded in ibiTreated μ-slides (3 × 10[4] cells/well; Ibidi) one day prior to infection. Cells were challenged with GFP-CA-labeled virus via

spinoculation (1200 × g for 1 h at 15 °C) in the presence of polybrene (10 μg/ml). Following spinoculation, the medium was replaced with prewarmed medium containing BioTracker 650 Red Nuclear Dye (0.5 μl/ml; Millipore). At 6 hours post-infection, the cells were fixed with 4% paraformaldehyde in PBS for 10 minutes and rinsed twice with PBS. Confocal z-slices at the equatorial plane of the cells were acquired. Each field of view (83 μm²) contained an average of 4 cells, and 8 fields were acquired per sample. A custom MATLAB program was used to quantify the efficiency of nuclear import of GFP signals[8]. First, the diffraction-limited GFP signals were detected, and their positions were determined in each image using Localize[76]. Second, nuclear masks were generated using the BioTracker 650 signal. Finally, the colocalization of each GFP spot with the nuclear masks was determined. The percentage of total GFP spots that colocalized with the nuclear masks was calculated.

### Integration site analysis

HeLa cells were seeded in 48-well plates (3 × 10⁴ cells/well) and challenged the next day with VSV-G-pseudotyped lentiviral particles containing the HIV-8.4 kb or HIV-3.1 kb genomes. For cells infected with the HIV-3.1 kb vector, medium containing 10 μM PF74, 10 nM LEN, 100 nM LEN, or DMSO (0.5%) was added 10 h after infection. Cells infected with the HIV-8.4 kb vector were left untreated or 10 μM PF74 was added 8 hours after infection. The next day, cells from multiple wells were pooled for each sample, cultured for 4 days, and the genomic DNA was extracted using QIAamp DNA Blood Maxi kit (Qiagen). Genomic DNA containing proviruses was enzymatically fragmented and A-tailed (NEBNext Ultra II FS; NEB), ligated to adapter linkers, PCR amplified, and sequenced using Illumina MiSeq as previously described[77,78]. The sequences of the host-viral junctions and the host DNA breakpoints were determined. Occasionally (~2% of the time), sequences with the same host-viral junction but different host DNA breakpoints were identified. These clones, which likely arose from cell division following the initial integration event or capturing of both the 5'LTR and 3'LTR of the same proviral genome, were treated as a single integration event. The host DNA sequence from each unique integration site was then mapped to the human genome (hg19). Integration sites from HIV-1-infected human PBMCs and random integration sites were obtained from computer-generated integration across the human genome (hg19) as previously reported[59]. Gene density was defined as the number of RefSeq hg19 genes within ± 500 kb of each integration site, with the mean density reported for each dataset. Chromosomal regions that lie within 500 nm of NSs are defined as SPADs, yielding 1,547,458 SPAD sequences each of 100 bp in length[79]. Chromosomal segments that were in close proximity to LaminB1 are defined as LADs[80], and were downloaded from the University of California, Santa Cruz (UCSC) Genome Browser[81].

### Data analysis and statistics

The Welch's unpaired t-test was used to analyze parametric data. Mann-Whitney U test was used to analyze nonparametric data. Chi-square tests were used to compare integration site frequencies; p values are reported in Supplementary Tables 2–5. All statistical tests were performed in Prism 10 (GraphPad Software).

### Reporting summary

Further information on research design is available in the Nature Portfolio Reporting Summary linked to this article.

## Data availability

The authors declare that all data supporting the findings of this study are available within the paper and its supplementary information files. Source data are provided with this paper.

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

## Acknowledgements

We thank members of the Pathak and Hu laboratories for valuable discussions and suggestions during manuscript preparation. The LADs and SPADS annotations were generously provided by Alan Engelman. This research was supported [in part] by the Intramural Research Program of the National Institutes of Health (NIH) (Z1A BC011436 to V.K.P. and Z1A BC010504 to W.-S.H.). Supplemental funding was provided by Office of AIDS Research (to V.K.P. and to W.-S.H.) to support collaborative interactions with the Behavior of HIV in Viral Environments Center (U54AI170855). S.C.P. and X.W. were supported by National Cancer Institute under contract no. 75N91019D00024. The contributions of the NIH author(s) were made as part of their official duties as NIH federal employees, are in compliance with agency policy requirements, and are considered Works of the United States Government. However, the findings and conclusions presented in this paper are those of the author(s) and do not necessarily reflect the views of the NIH or the U.S. Department of Health and Human Services.

## Author contributions

R.C.B. and V.K.P. conceived and supervised the project and designed the experiments. R.C.B., W.S-H. and V.K.P. and interpreted the results with input from all authors. R.C.B. performed the experiments. S.C.P., S.H.H. and X.W. performed and analyzed the integration site studies. R.S. and K.D.-F. generated the HeLa: CKO cell line. E.B. generated the pHmNG-iHALO and pMLV-HALO-SRSF1-IRES-Hygro constructs. O.A.N. provided the pLR2-Vpr-IN-sfGFP construct. R.C.B. and V.K.P. wrote the manuscript. All authors edited and approved the manuscript.

## Competing interests

The authors declare no competing interests.
