## [Transparent Peer Review file · Nature Communications]

HIV-1 uncoating location dictates sites of integration

Corresponding Author: Dr Vinay Pathak

Version 0:

Reviewer comments:

Reviewer #1

(Remarks to the Author)

This study from Burdick and colleagues used live and fixed cell imaging to understand the impact of CA inhibitors, and CPSF6 binding, on nuclear trafficking of HIV-1 and the staging of the capsid disassembly process. The approaches used have been largely established by the authors previously, although in some cases different labelling strategies are employed to accomplish the goals of the study. Broadly, this study is an impressive body of work and provides insight into how capsid inhibitors influences these critical steps of infection. Most interesting is the differential effect, observed in some cases, between PF75 and Lenacapavir, which generally aligns with previous studies of these drugs and provides in cellulose appreciation of their subtly different mechanisms of action. I do note some issues that I think should be resolved below, but these generally fall into the area of data rollout and interpretation in one case. Realizing there are constraints associated with the number of panels one can put in an individual figure, there is a lot of data in the extended figures that is as important or, in my opinion, more important, than data that is provided in the actual figures. This makes the manuscript somewhat difficult to consume and appreciate. I also have concerns about the imaging experiments utilizing IN-sfGFP, as described below, and feel like these experiments do not add much to the manuscript and are likely to leave the average reader unfamiliar with the nuances of these imaging approaches confused or likely to infer things that I do not think are true, based on the data provided. I also think it would be worthwhile to include a few files of the actual live cell movies utilized in the analysis, realizing that there are far too many movies to include every relevant movie. Examples of each approach would nevertheless be useful and facilitate enjoyable consumption of this study. I think resolution of these issues will lead to a stronger manuscript that I would be supportive of publication in a high-profile journal such as Nature Communications.

Supp Fig 1:

A: Although HeLa cells are heterogenous, the speckles in the figure look a bit irregular. Does expression of the SRSF1 construct impact the nuclear localization of SRRM2? A cell stained with SRRM2 that was not transduced with the HALO tagged version of the panel

E looks at infectivity of unlabeled vs dual-labelled virus. Does unlabeled mean virus mean virus with neither the GFP-CA or Halo tag (ie just wt gag), or perhaps the ligand is not added? Given the number of things this could mean, please clarify. All version of virus containing an engineered construct should be compared to a virus with just wt gag and no engineered construct, if this is not what is actually presented.

Suppl Fig 2: The authors attribute differences in Sup Fig 2B and Fig 1d to “phototoxicity from higher frame rates and exposure to 3 lasers”. First, do the authors mean photobleaching? In either case, this commentary feels more appropriate in the results text, rather than the supplemental figure legend.

Is it possible to run statistics on Sup Fig 2D. With n of 35 or 48, how many viral particles contribute to this difference?

E: can an allusion to PF74 treatment be made in the actual figure. It might be useful to plot both wt and PF74 treatment in a single graph so the reader can appreciate the differences.

Fig 1: The data seem clear and well analyzed. One might argue that 33 virions isn't a huge data set, but this reviewer recognized these experiments are hard. On that front though, it might aid interpretation to see, at least in a supplemental figure, the intensities of those virions (138 of them) that did not lose content marker or GFP-CA during the 10-12 hour period, as this would reveal the degree to which photobleaching did not occur, and could even be used to normalize the data in cores which did experience cmHALO loss and GFP-CA loss as necessary.

The authors state “A minority (15%) of viral cores exited the NS following core rupture and substantial loss of CA”. This isn't entirely clear, but I think the authors mean “following core rupture but BEFORE/PRIOR TO substantial loss of CA”, based on the preceding sentence. Please confirm and/or clarify.

Is 1F simply the reciprocal of 1E?

Fig 2: Reasonably straightforward, no concerns.

Fig 3: Same concerns as for Sup Fig 4 below. I'm not sure what this data means, but I am concerned that readers will infer meaning from these data that is not supported by the data, particularly the data with Ral in Sup Fig 4 panel F.

Extended Data Fig 4: I have concerns about this figure and its interpretation, specifically with the utility of the IN-sfGFP construct. It seems to me that the IN-sfGFP construct is not ideally suited for the analysis intended, as the viral core will likely contain many more IN-sfGFP molecules than end up associated with the PIC after core disassembly. The virus also contributes native integrase, which may also associate with the PIC, perhaps more efficiently. This would explain the lack of correlation between IN-sfGFP signal and integration, in panel F. As such, signal loss may represent a secondary readout of capsid disassembly, in as much as signal loss is not associated with integration, as was presumably desired. Finally, in panel G, it would again be important to understand what unlabeled means, and compare the infectivity to native virions.

Extended Data Fig 5/ Fig 4: These are strong data, but as with the other rollout of the data, one gets a bit of whiplash going to the supplemental figures, which contain data that is critical and does not seem to feel supplemental, and the actual figure. I realize there are only so many panels one can add to a figure without being cluttered or just in poor taste, but in this case, I would consider moving some of the key data in the extended fig into Fig 4, (IMO C and H) and making the integration data its own figure. Statistical analysis of panels currently listed in g-j are needed, with the understanding that even small differences in integration sites are typically significant in such studies.

Reviewer #2

(Remarks to the Author)

The manuscript by Burdick et al. presents a technically sophisticated and conceptually compelling study that elucidates how the spatial location of HIV-1 uncoating within nuclear speckles (NSs) influences integration site selection. Based on the results of live-cell imaging and dual-labeling strategies, and by pharmacological targeting of capsid integrity, the authors suggest that the vDNA integrates near the sites of uncoating and that capsid disassembly within NSs promotes integration into gene-rich, transcriptionally active speckle-associated domains (SPADs), whereas pharmacological disruption of capsid integrity, affects the localization of CA in NS, and redirects integration toward gene-poor lamina-associated domains (LADs). Furthermore, their studies highlight the role of CPSF6 interaction as a bridge between CA and NS for positioning these capsids within the NS.

The studies are based on the foundation of prior studies. However, the weakness is that the results presented are based on studies using very high concentration of CA-inhibitors (~1000 to 100 times more than the EC50) and the effects on integration target site selection using full length vectors are marginal, questioning the significance of these studies. The following highlights the major and minor weaknesses.

Major weaknesses:

1) The current hypothesis is that exit of capsid from NS leads to integration into nearest LADs. This hypothesis is based on the imaging and independently by sequence analysis of integration sites using different vectors. In the sequence analysis, while the CA inhibitors reduce integration into SPADs, it will be important to demonstrate this in the imaging. The markers for SPADs (e.g., SON, SC35) and LADs (e.g., Lamin B1) can be co-labeled to validate integration site proximity in the presence and absence of CA inhibitors.

2) LEDGF/p75 is a key host factor that guides HIV-1 integration into active chromatin. It is unclear whether PICs resulting from inhibitor-induced uncoating retain LEDGF/p75 association or whether integration into LADs reflects altered host factor engagement.

Analyzing LEDGF association with PICs in the context of NS exit will shed more light on the mechanism.

3) What is the consequence of integration of virus into LADs? The manuscript shows altered integration site distribution. But what is the consequence of these integrations on replication of the virus? The authors should measure viral gene expression and replication competence from LAD-targeted proviruses.

4) The fact that Bgl-mCherry containing HIV-3.9kb vector showed 7-fold increase in number of cells containing transcription site in the presence of PF74 (Extended Data Fig. 5c) suggests that there could be an increase in transcriptional activity. This needs further explanation, as there was no significant difference in the association with NS in the presence and absence of CA-inhibitors.

5) The authors note that IN-sfGFP signal loss is not strictly correlated with integration. Even in the presence of Integration inhibitor RAL the IN-sfGFP signal loss is observed. Explanations (e.g., proteolysis, nuclear export) for this is not discussed.

6) CPSF6 is shown to retain viral cores within NSs, but the domain-specific contributions (e.g., FG motif, phase separation domains) are not dissected. How does CPSF6 structurally mediate NS retention and integration fidelity. Use of CPSF6 mutants lacking the FG motif or phase separation domains in CPSF6-KO cells and assessing NS localization, core retention, and integration targeting to determine which domains are essential for NS anchoring and SPAD-directed integration.

Minor Points

1. The manuscript would benefit from a deeper discussion of NSs as dynamic transcriptional condensates rather than passive chromatin landmarks. Recent literature on NS phase separation and transcriptional activity (e.g., Ilik et al., *Elife* 2020; Jang et al., *NAR* 2024) could help contextualize their role in integration targeting.

2. While the differential kinetics of PF74 and LEN are well-characterized, the authors might elaborate on whether differences in binding affinity or FG-pocket occupancy contribute to the observed disparities.

3. The persistence of CA signal post-uncoating and its association with IN-sfGFP is intriguing. A brief discussion of how this relates to current models of intasome assembly or nuclear PIC stabilization would enrich the mechanistic narrative.

Reviewer #3

(Remarks to the Author)

Version 1:

Reviewer comments:

Reviewer #1

(Remarks to the Author)

I feel the authors have responded reasonably to the concerns and suggestions I provided previously. I now feel that the conclusions are supported by the data and the manuscript itself is more easily consumed, particularly so for people not directly involved in this particular area of research, which I think is important in a journal such as this. I have no further concerns and appreciate the authors responses to my comments.

Reviewer #2

(Remarks to the Author)

The revisions made in the manuscript have ironed out many of the creases nicely. Particularly, the re-arrangement of some figures, bringing some portions of the extended data into the main figures, has improved the readability of the narrative. The additions made in the discussion part has also improved the contextuality of the narrative in the light of the existing literature. Some of the questions such as those related to association of LEDGF in PICS in the presence of CA inhibitors, the differential kinetics of PF74 and LEN, increased number of transcription sites observed in HIV-3.9kb vector etc... remain unresolved. However, remaining questions have been answered diligently.

Reviewer #3

(Remarks to the Author)

Below, we provide point-by-point responses to the reviewer's comments (blue font).

Reviewer #1 (Remarks to the Author)

This study from Burdick and colleagues used live and fixed cell imaging to understand the impact of CA inhibitors, and CPSF6 binding, on nuclear trafficking of HIV-1 and the staging of the capsid disassembly process. The approaches used have been largely established by the authors previously, although in some cases different labelling strategies are employed to accomplish the goals of the study. Broadly, this study is an impressive body of work and provides insight into how capsid inhibitors influences these critical steps of infection. Most interesting is the differential effect, observed in some cases, between PF75 and Lenacapavir, which generally aligns with previous studies of these drugs and provides in cellulose appreciation of their subtly different mechanisms of action. I do note some issues that I think should be resolved below, but these generally fall into the area of data rollout and interpretation in one case. Realizing there are constraints associated with the number of panels one can put in an individual figure, there is a lot of data in the extended figures that is as important or, in my opinion, more important, than data that is provided in the actual figures. This makes the manuscript somewhat difficult to consume and appreciate.

Response: We thank the reviewer for the kind and supportive comments. We agree that that the abbreviated text and frequent cross-referencing between main figures and supplementary materials reduced the overall readability of the manuscript. In response, we have substantially reorganized the figures: the original Fig. 1 has been split into two new figures, the original Fig. 4 has also been divided into two new figures, and a significant portion of the supplementary material has been incorporated into the main figures. Collectively, we believe these changes reduce the back-and-forth between the main figures and the Extended Data figures and have improved the clarity and readability of the manuscript.

I also have concerns about the imaging experiments utilizing IN-sfGFP, as described below, and feel like these experiments do not add much to the manuscript and are likely to leave the average reader unfamiliar with the nuances of these imaging approaches confused or likely to infer things that I do not think are true, based on the data provided.

Response: We agree with the reviewer's concern regarding the potential for confusion in interpreting the IN-sfGFP imaging experiments. The data presented in the original submission showed that the IN-sfGFP dissociates from the PIC ~14 minutes after uncoating and does not remain associated with the PIC until the time of integration (original Extended data Fig. 4). While these data contribute to the characterization of IN-sfGFP as a tool for labeling PICs, we agree that they do not add to the main conclusions of our study and have removed them in the resubmitted manuscript to avoid confusion.

Despite these limitations, IN-sfGFP labeling allowed us to transiently track viral complexes after NS exit and to demonstrate that low levels of CA remain associated with these complexes immediately after NS exit and substantial CA loss indicating uncoating. We believe this is a valuable observation that would have been difficult to obtain without the additional IN-sfGFP marker and have retained this data in the revised Fig. 4.

I also think it would be worthwhile to include a few files of the actual live cell movies utilized in the analysis, realizing that there are far too many movies to include every relevant movie. Examples of each approach would nevertheless be useful and facilitate enjoyable consumption of

this study. I think resolution of these issues will lead to a stronger manuscript that I would be supportive of publication in a high-profile journal such as Nature Communications.

Response: We agree and have provided movies for each example shown in the main figures (10 movies total), along with a detailed description in the Methods section describing how the movies were generated.

Supp Fig 1:

A: Although HeLa cells are heterogenous, the speckles in the figure look a bit irregular. Does expression of the SRSF1 construct impact the nuclear localization of SRRM2?

Response: We observe no obvious differences in the morphology of nuclear speckles in cells expressing HALO-SRSF1. We have now included an example image of unmodified HeLa cells stained with the anti-SRRM2 antibody (Fig. 1a).

A cell stained with SRRM2 that was not transduced with the HALO tagged version of the panel E looks at infectivity of unlabeled vs dual-labelled virus. Does unlabeled mean virus with neither the GFP-CA or Halo tag (ie just wt gag), or perhaps the ligand is not added? Given the number of things this could mean, please clarify. All version of virus containing an engineered construct should be compared to a virus with just wt gag and no engineered construct, if this is not what is actually presented.

Response: We agree that this was unclear. The virus contains only wild-type Gag and has now been renamed “WT” (Extended Data Fig. 1c). The Western blot confirms the absence of GFP-CA or HALO proteins in these virions, indicating that they are truly unlabeled (Extended Data Fig. 1b).

Suppl Fig 2: The authors attribute differences in Sup Fig 2B and Fig 1d to “phototoxicity from higher frame rates and exposure to 3 lasers”. First, do the authors mean photobleaching? In either case, this commentary feels more appropriate in the results text, rather than the supplemental figure legend.

Response: This explanation was originally included to account for the slightly higher-than-expected proportion of particles that disappeared in some shorter movies under DMSO control conditions corresponding to the experiments shown in new Fig. 2 and Extended Data Fig. 2. However, we agree that this commentary is unnecessary and potentially distracting. Accordingly, we have removed it from the revised manuscript.

Is it possible to run statistics on Sup Fig 2D. With n of 35 or 48, how many viral particles contribute to this difference?

Response: The difference between DMSO (4 of 35; ~11%) and PF74 (18 of 48; ~38%) is statistically significant (Fisher’s exact test, $P = 0.006$). We have added this point to the results section (Pg. 6, lines 6-9). For simplicity, we now present pie charts showing the percentages of viral cores that lost both cmHALO and GFP-CA, lost only cmHALO, or retained both cmHALO and GFP-CA (Extended Data Fig. 2a).

E: can an allusion to PF74 treatment be made in the actual figure. It might be useful to plot both wt and PF74 treatment in a single graph so the reader can appreciate the differences.

Response: We have generated a new graph comparing the kinetics of cmHALO and GFP-CA loss in untreated control cells and PF74-treated cells (Extended Data Fig. 2c).

Fig 1: The data seem clear and well analyzed. One might argue that 33 virions isn't a huge data set, but this reviewer recognized these experiments are hard. On that front though, it might aid interpretation to see, at least in a supplemental figure, the intensities of those virions (138 of them) that did not lose content marker or GFP-CA during the 10-12 hour period, as this would reveal the degree to which photobleaching did not occur, and could even be used to normalize the data in cores which did experience cmHALO loss and GFP-CA loss as necessary.

Response: We thank the reviewer for noting that obtaining 33 virions that uncoated in the nucleus required tracking 170 nuclear viral cores. The reviewer also raises a good point about obtaining fluorescence intensities of viral cores that did not uncoat. We tracked and quantified the fluorescence intensities of 30 randomly selected particles that did not uncoat, which were sufficient to quantify photobleaching. The data, now included in Extended Data Fig. 1e, show minimal photobleaching over the 2-hour imaging period.

In order to simplify the analysis, we now focus on only the 31 viral cores that lost both cmHALO and GFP-CA during the movie (rather than the 33 viral cores, two of which lost only the cmHALO signal).

The authors state "A minority (15%) of viral cores exited the NS following core rupture and substantial loss of CA". This isn't entirely clear, but I think the authors mean "following core rupture but BEFORE/PRIOR TO substantial loss of CA", based on the preceding sentence. Please confirm and/or clarify.

Response: We apologize for the confusion. Our observation was that, in some cases, a small amount of GFP-CA signal could be reliably detected exiting the NS following rupture and substantial GFP-CA loss within the speckle. Perhaps the original wording was unclear because the viral complex that exits the nuclear speckle should no longer be considered a viral core, which implies an intact or mostly intact capsid shell. We have now clarified this point in the manuscript (Pg. 5, lines 18-21), where we state: "A minority (16%) of viral reverse transcription/preintegration complexes (RTCs/PICs) exited the NSs following loss of cmHALO (core rupture) and substantial loss of GFP-CA (uncoating) and subsequently lost all detectable GFP-CA signal outside the NSs (Fig. 1f; example in Fig. 1h)."

Is 1F simply the reciprocal of 1E?

Response: Not necessarily, since some GFP-CA puncta could have remained within NSs during the entire observation period.

Fig 2: Reasonably straightforward, no concerns.

Response: Thanks! We would also like to emphasize that we have modified all figures to minimize switching between the main figures and the supplementary material, with key data now incorporated into the main figures where possible.

Fig 3: Same concerns as for Sup Fig 4 below. I'm not sure what this data means, but I am concerned that readers will infer meaning from these data that is not supported by the data, particularly the data with Ral in Sup Fig 4 panel F.

Response: As discussed above, we have removed these data from the revised manuscript and now use IN-sfGFP only for short-term tracking of viral particles after uncoating, enabling us to

quantify the small amount of CA that remains associated with viral complexes following uncoating.

Extended Data Fig 4: I have concerns about this figure and its interpretation, specifically with the utility of the IN-sfGFP construct. It seems to me that the IN-sfGFP construct is not ideally suited for the analysis intended, as the viral core will likely contain many more IN-sfGFP molecules than end up associated with the PIC after core disassembly. The virus also contributes native integrase, which may also associate with the PIC, perhaps more efficiently. This would explain the lack of correlation between IN-sfGFP signal and integration, in panel F. As such, signal loss may represent a secondary readout of capsid disassembly, in as much as signal loss is not associated with integration, as was presumably desired.

Response: We agree with the reviewer's interpretation and believe this explanation is entirely consistent with our observations. We hypothesize that fluorescent protein-tagged IN does not stably associate with the PIC. Specifically, although the viral core contains many IN-sfGFP molecules prior to uncoating, only a subset of native integrase is expected to be incorporated into the functional intasome. Our data suggest that IN-sfGFP is largely excluded from this complex and instead diffuses away from the PIC (DNA and/or protein) following capsid disassembly, consistent with the lack of correlation between IN-sfGFP signal and integration. As discussed above, we have only used IN-sfGFP signal to track viral complexes for a short time period after uncoating to quantify the small amount of CA that remains associated with the PIC.

Finally, in panel G, it would again be important to understand what unlabeled means, and compare the infectivity to native virions.

Response: As discussed above, this data has now been removed from the revised manuscript.

Extended Data Fig 5/ Fig 4: These are strong data, but as with the other rollout of the data, one gets a bit of whiplash going to the supplemental figures, which contain data that is critical and does not seem to feel supplemental, and the actual figure. I realize there are only so many panels one can add to a figure without being cluttered or just in poor taste, but in this case, I would consider moving some of the key data in the extended fig into Fig 4, (IMO C and H) and making the integration data its own figure. Statistical analysis of panels currently listed in g-j are needed, with the understanding that even small differences in integration sites are typically significant in such studies.

Response: We agree and have split the original Fig. 4 into two new figures (Fig. 5 and 6), moving much of the supplemental data into the main figures. We have also added statistical analyses for key comparisons to the bar graphs and now emphasize in the Results (Pg. 9, lines 30-31) and in the figure legend that all statistical comparisons are provided in Supplementary Tables S2–S5.

Reviewer #2 (Remarks to the Author)

The manuscript by Burdick et al. presents a technically sophisticated and conceptually compelling study that elucidates how the spatial location of HIV-1 uncoating within nuclear speckles (NSs) influences integration site selection. Based on the results of live-cell imaging and dual-labeling strategies, and by pharmacological targeting of capsid integrity, the authors suggest that the vDNA integrates near the sites of uncoating and that capsid disassembly within NSs promotes integration into gene-rich, transcriptionally active speckle-associated domains

(SPADs), whereas pharmacological disruption of capsid integrity, affects the localization of CA in NS, and redirects integration toward gene-poor lamina-associated domains (LADs). Furthermore, their studies highlight the role of CPSF6 interaction as a bridge between CA and NS for positioning these capsids within the NS.

The studies are based on the foundation of prior studies. However, the weakness is that the results presented are based on studies using very high concentration of CA-inhibitors (~1000 to 100 times more than the EC₅₀) and the effects on integration target site selection using full length vectors are marginal, questioning the significance of these studies. The following highlights the major and minor weaknesses.

Response: We thank the reviewer for this thoughtful critique but respectfully disagree with the conclusion that the data question the significance of these studies. It is important to note that we used capsid inhibitors as experimental tools to induce uncoating at specific times to interrogate the relation between uncoating location, CPSF6-dependent NS localization, and integration site selection. We deliberately used high concentrations of capsid inhibitors to ensure that the majority of nuclear capsids underwent rupture and uncoating with comparable kinetics across conditions. As described in the text (Pg. 5, lines 26-28), we selected 100 nM lenacapavir (LEN; ~2,500× EC₅₀) because the kinetics of capsid integrity loss at this concentration closely matched those observed with 10 μM PF74 (~100× EC₅₀). Accordingly, we were careful not to imply physiological relevance of these concentrations and explicitly state in the Discussion (Pg. 12, lines 25-26) that PF74 and LEN potently inhibit nuclear import, making it unlikely that nuclear capsids would encounter such inhibitor levels in vivo. We also compare our findings with studies in which capsid inhibitors were added at the time of infection, a scenario more relevant to treated individuals. Although we are not trying to predict what happens in patients treated with LEN, individuals receiving single subcutaneous dose of LEN have plasma concentrations from 10 nM to 100 nM range (Link et al., Nature 2020; PMID: 32612233).

With respect to the more modest shift in integration site distribution observed with the near full-length virus, this outcome was expected and reflects a mixture of integration events arising from natural, reverse transcription-dependent uncoating inside NSs occurring prior to inhibitor addition, and inhibitor-induced uncoating outside NSs. Because capsid inhibitor addition cannot be perfectly timed—early addition blocks reverse transcription, whereas late addition permits natural uncoating—it is not possible to force all integration events from the near full-length virus to arise exclusively from inhibitor-induced uncoating.

To balance these constraints, we used prior time-of-addition experiments to select 8 hours post-infection as the optimal window to assess integration following inhibitor-induced uncoating. This limitation highlights the utility of the short vectors used in our study: because viral cores containing short genomes uncoat inefficiently, integration is minimal in the absence of capsid inhibitors. As a result, the majority of integration events observed with short vectors can be confidently attributed to inhibitor-induced uncoating, explaining the more pronounced shift in integration site distribution relative to the near full-length virus.

We apologize if these points were not sufficiently clear and believe the revised manuscript now better articulates both the rationale for our experimental design and the appropriate interpretation of the results.

Major weaknesses:

1) The current hypothesis is that exit of capsid from NS leads to integration into nearest LADS. This hypothesis is based on the imaging and independently by sequence analysis of integration

sites using different vectors. In the sequence analysis, while the CA inhibitors reduce integration into SPADS, it will be important demonstrate this in the imaging. The markers for SPADs (e.g., SON, SC35) and LADs (e.g., Lamin B1) can be co-labeled to validate integration site proximity in the presence and absence of CA inhibitors.

Response: We appreciate the reviewer's suggestion to further validate integration site proximity by imaging. However, it is important to note that SPADs are defined as chromatin regions proximal to NSs but are not expected to directly colocalize with NS markers such as SON or SC35. Thus, reduced integration within SPADs would not be predicted to manifest as decreased colocalization with NS markers per se.

Instead, we addressed this question by quantifying the spatial positioning of proviruses relative to both NSs and the nuclear periphery. In the previous Fig. 4e (now Fig. 5i), we show that proviruses resulting from capsid inhibitor-induced uncoating are positioned $\sim 0.4 \mu\text{m}$ farther from NSs than proviruses arising from natural uncoating within NSs. In parallel, in the previous Fig. 4f (now Fig. 5j), we demonstrate that these proviruses are located a similar distance farther from the nuclear periphery. In these experiments, NSs were identified by immunofluorescence staining of SRRM2, an essential NS protein (Ilik et al., 2020 eLife; PMID: 33095160), and the nuclear envelope was localized by the sharp reduction in the diffuse Bgl-mCherry signal at the nuclear edge, which we previously showed colocalizes with nuclear pore proteins (Burdick et al., 2020 PNAS; PMID: 32094182).

Consistent with these imaging-based spatial shifts, integration site analysis reveals a decrease in integration within SPADs and a modest increase in integration within LADs following capsid inhibitor treatment. Together, these complementary imaging and sequencing data support a model in which capsid inhibitor-induced exit from NSs leads to integration in chromatin regions located farther from NSs, including LADs. We have clarified this interpretation in the revised text (Pg. 11, lines 8-24), and we hope that the reorganization of the figures and text makes this conclusion clear.

2) LEDGF/p75 is a key host factor that guides HIV-1 integration into active chromatin. It is unclear whether PICs resulting from inhibitor-induced uncoating retain LEDGF/p75 association or whether integration into LADs reflects altered host factor engagement. Analyzing LEDGF association with PICs in the context of NS exit will shed more light on the mechanism.

Response: We completely agree that LEDGF/p75 is a key host factor guiding HIV-1 integration into active chromatin, and that its engagement with PICs following inhibitor-induced uncoating represents an important mechanistic question. We believe, however, that a definitive analysis of LEDGF-PIC interactions will require the development of new tools that enable direct tracking of PICs from uncoating through the time of integration. As noted above, IN-sfGFP is not well suited for this purpose, as it dissociates from PICs shortly after uncoating, even when integration is blocked by raltegravir. Nonetheless, the data presented in the current study support the conclusion that the location of uncoating influences the selection of integration sites. Whether these integration events are LEDGF-dependent or LEDGF-independent is therefore an important question that we believe will be most effectively addressed in future studies using approaches specifically designed to track PICs from uncoating to integration.

3) What is the consequence of integration of virus into LADs? The manuscript shows altered integration site distribution. But what is the consequence of these integrations on replication of

the virus? The authors should measure viral gene expression and replication competence from LAD-targeted proviruses.

Response: The reviewer raises an important point regarding the functional consequences of viral integration into LADs. In our previous work (Burdick et al., 2020 PNAS), we showed that CA mutants that integrate almost exclusively into LADs near the nuclear envelope do not exhibit reduced viral infectivity under standard cell culture conditions. These findings are consistent with prior studies examining the CPSF6-binding capsid mutants A77V (Saito et al., 2016 J Virol; PMID 27307565) and N74D (Lee et al., 2010 CHM: PMID: 20227665), which also did not display reduced infectivity despite altered integration site distributions. Together, these observations suggest that integration into LADs does not necessarily impair proviral gene expression or viral transcription in vitro. Notably, the CPSF6-binding capsid mutant A77V reverted to wild type in a humanized mouse model (Saito et al., 2016 J Virol; PMID 27307565), suggesting that the impact of integration site selection on viral replication is biologically important and may not be fully captured in cell culture systems. Thus, while our current study focuses on the relationship between the uncoating location and integration site selection, the relationship between integration into LADs and long-term viral gene expression or replication fitness in vivo remains an open and important question that we hope to address in future studies.

4) The fact that Bgl-mCherry containing HIV-3.9kb vector showed 7-fold increase in number of cells containing transcription site in the presence of PF74 (Extended Data Fig. 5c) suggests that there could be an increase in transcriptional activity. This needs further explanation, as there was no significant difference in the association with NS in the presence and absence of CA-inhibitors.

Response: While it would indeed be interesting if proviruses resulting from PF74-induced uncoating exhibited increased transcriptional activity as a consequence of altered integration site selection, we believe there is a simpler explanation for the observation in original Extended Data Fig. 5c (now Fig. 5b). We previously showed that HIV-1-based vectors with small genomes uncoat inefficiently and, as a result, fail to integrate and generate transcriptionally active proviruses (Burdick et al., 2024 Sci Adv; PMID: 38657061). Treatment with PF74 induces capsid disassembly, thereby permitting these small vector genomes to integrate more efficiently and produce transcriptionally active proviruses. Accordingly, the observed increase in the number of transcription sites directly reflects an increase in successful integration events rather than enhanced transcriptional activity per provirus. We have now clarified this point in the manuscript (Pg. 8, lines 21-25).

Regarding the lack of a significant difference in NS association in the presence or absence of CA inhibitors, very few proviruses colocalize with NSs under any condition (original Extended Data Fig. 5c, now Fig. 5h), consistent with the fact that NSs are located in interchromatin space. Instead, we quantified the distance between proviruses and the edge of the nearest NS (original Fig. 4e, now Fig. 5i). These analyses show that WT proviruses are located $\sim 0.6 \mu\text{m}$ from the nearest NS, whereas proviruses derived from short vectors in cells treated with PF74 or LEN are positioned further away, at $\sim 1.0 \mu\text{m}$ from the nearest NS. Thus, CA inhibitor treatment alters integration positioning relative to NSs.

5) The authors note that IN-sfGFP signal loss is not strictly correlated with integration. Even in the presence of Integration inhibitor RAL the IN-sfGFP signal loss is observed. Explanations (e.g., proteolysis, nuclear export) for this is not discussed.

Response: In the revised text and in our response to Reviewer 2, we have emphasized the key insights gained from using the IN-sfGFP label and the limitations of this approach. There are two possible explanations for the lack of correlation between IN-sfGFP signal loss and integration. First, although IN-sfGFP is packaged into virions, it may not be incorporated into the integrase complex associated with viral DNA ends. Instead, most or all of the IN-sfGFP likely dissociates from the viral preintegration complex upon capsid disassembly. Alternatively, a small amount of IN-sfGFP may be incorporated into the integrase complex, but the resulting signal may be below the limit of detection; in either case, the detectable IN-sfGFP is lost upon uncoating.

Our data suggests that IN-sfGFP dissociates and diffuses away from the viral complex before integration; we speculate that the sfGFP tag interferes with stable association of IN with the PIC, which limits its utility as a marker for tracking preintegration complexes until integration.

6) CPSF6 is shown to retain viral cores within NSs, but the domain-specific contributions (e.g., FG motif, phase separation domains) are not dissected. How does CPSF6 structurally mediate NS retention and integration fidelity. Use of CPSF6 mutants lacking the FG motif or phase separation domains in CPSF6-KO cells and assessing NS localization, core retention, and integration targeting to determine which domains are essential for NS anchoring and SPAD-directed integration.

Response: Previous studies have shown that both the FG motif and the mixed-charge domain of CPSF6 are essential for nuclear import of viral cores (Jang et al., 2024 NAR; PMID: 39258548). As a result, the effects of these mutations on viral core localization to NSs cannot be directly assessed because the viral cores do not enter the nucleus, making this a technically challenging question to address. Interestingly, CPSF6 mutants have been described in which the C-terminal mixed-charge domain is functionally replaced with alternative nuclear localization signals (Rohlfes et al. 2025 PLoS Pathog; PMID: 39823525). Our future goal is to leverage such approaches to disentangle CPSF6's role in nuclear import versus NS localization. However, given the technical complexity of these experiments, we believe that addressing these questions in depth is best suited for future studies.

Minor Points

1. The manuscript would benefit from a deeper discussion of NSs as dynamic transcriptional condensates rather than passive chromatin landmarks. Recent literature on NS phase separation and transcriptional activity (e.g., Ilik et al., Elife 2020; Jang et al., NAR 2024) could help contextualize their role in integration targeting.

Response: We agree with the reviewer and have revised the Introduction to better describe nuclear speckles as dynamic, transcriptionally active condensates rather than passive chromatin landmarks, incorporating relevant recent literature to provide additional context for our findings (Pg. 2, starting at line 29).

2. While the differential kinetics of PF74 and LEN are well-characterized, the authors might elaborate on whether differences in binding affinity or FG-pocket occupancy contribute to the observed disparities.

Response: The reasons for different kinetics of NS exit with PF74 and LEN are not well understood. We speculated in the Discussion that it likely involves the FG-binding pocket. On Pg. 12, lines 1-6, we state: "The basis for these kinetic differences in the loss of capsid-

associated CPSF6 remains unclear but may involve greater accessibility or higher occupancy of FG-binding pocket by PF74 compared to LEN. Despite their opposing effects on capsid stability – PF74 promoting capsid disassembly and LEN stabilizing broken capsids – both treatments ultimately led to vDNA being released outside NSs, as evidenced by similar provirus localization and integration site patterns.”

3. The persistence of CA signal post-uncoating and its association with IN-sfGFP is intriguing. A brief discussion of how this relates to current models of intasome assembly or nuclear PIC stabilization would enrich the mechanistic narrative.

Response: We agree that this is an intriguing observation that was made possible by IN-sfGFP labeling of viral complexes. At present, however, we lack experimental evidence that would provide mechanistic insight into the persistence of CA signal post-uncoating. Therefore, although this finding is interesting, we believe it would be premature to speculate on potential roles for CA in intasome formation or integration.

Instead, we chose to frame this observation in the context of broader unresolved questions in the field. As noted in the Discussion (Pg. 13, lines 18-22), “Several key questions remain unresolved, including when and where intasome assembly occurs, when the intasome first encounters LEDGF/p75, and what mechanisms guide the intasome to its chromatin target. It also remains unclear whether the small amount of capsid that remains associated with the vDNA upon NS exit is a critical component of the PIC, as previously proposed.”

Reviewer #3 (Remarks to the Author)

Response: Thanks for reviewing our manuscript!